# Unsupervised Multi-object Segmentation by Predicting Probable Motion Patterns

**Laurynas Karazija**[*], **Subhabrata Choudhury**[*],
**Iro Laina, Christian Rupprecht, Andrea Vedaldi**

Visual Geometry Group
University of Oxford
Oxford, UK
{laurynas,subha,iro,chrisr,vedaldi}@robots.ox.ac.uk

## Abstract

We propose a new approach to learn to segment multiple image objects without manual supervision. The method can extract objects form still images, but uses videos for supervision. While prior works have considered motion for segmentation, a key insight is that, while motion can be used to identify objects, not all objects are necessarily in motion: the absence of motion does not imply the absence of objects. Hence, our model learns to predict image regions that are likely to contain motion patterns characteristic of objects moving rigidly. It does not predict *specific* motion, which cannot be done unambiguously from a still image, but a distribution of possible motions, which includes the possibility that an object does not move at all. We demonstrate the advantage of this approach over its deterministic counterpart and show state-of-the-art unsupervised object segmentation performance on simulated and real-world benchmarks, surpassing methods that use motion even at test time. As our approach is applicable to variety of network architectures that segment the scenes, we also apply it to existing image reconstruction-based models showing drastic improvement. Project page and code: https://www.robots.ox.ac.uk/~vgg/research/ppmp.

## 1 Introduction

Humans have an innate ability to segment individual objects in a picture, but learning this capability with an algorithm usually relies on manual supervision. In this paper, we consider the problem of learning to segment objects from visual data only — without externally provided labels. Algorithms for this task usually assume that objects are seen in different configurations and in front of different backgrounds. They then exploit cues such as the visual consistency and the co-occurrence of characteristic object parts to learn to discover and segment individual object instances.

Most such methods use still images as input and train with a reconstruction objective [19, 23, 37]. They work well on simple synthetic scenes, but they struggle in scenes with more complex visual appearance [27]. This has motivated the development of algorithms that use videos as input and can thus observe the motion of the objects as evidence of their presence. A common way of using motion for unsupervised learning is to seek for a compact representation to *reconstruct* the video itself [24, 26, 33, 53]. Effectively, such methods seek for a compressed representation of appearance, but do not sidestep entirely the difficult task of modeling it. This has motivated authors to look instead at reconstructing the video's *optical flow* [28, 57]. In fact, the optical flow measures the motion of the objects directly and is much simpler to model than the objects' appearance.

---

[*]Authors contributed equally.

In this paper, we propose a method that lies in-between these two classes, *i.e.* single image-based and video-based. Our model learns to segment objects in still images, and is thus based on appearance, but learns to do so using *video as a learning signal*, in an unsupervised manner. The learning process can be summarized as follows. Given an image, each pixel is assigned to a slot that represents a certain object. The quality of the assignments is then measured by the coherence of the (unobserved) optical flow within the extracted regions. Because predicting optical flow from a still image is intrinsically ambiguous, the method models *distributions* of probable flow patterns within each region. The idea is that (rigid) objects generate characteristic flow patterns that can be used to distinguish between them.

Note that, because the segmentation network is based on a single image, it will learn to partition all objects contained in the scene, not just the ones that actually move in the video, *i.e.* solving an object instance segmentation problem, rather than *motion* segmentation.

We derive closed-form distributions for the flow field generated by rigid objects moving in the scene. We also derive efficient expressions for the calculation of the flow probability under such models. The problem of decomposing the image into a number of regions is cast as a standard image segmentation task and an off-the-shelf neural network can be used for it.

As our method uses videos to train an image-based model, we introduce two new datasets which are straightforward video extensions of the existing image datasets CLEVR [25] and CLEVRTEX [27]. These datasets are built by animating the objects with initial velocities and using a physics simulation to generate realistic object movement. Our datasets are constructed with the realistic assumption that not all objects are moving at all times. This means that motion alone cannot be used as the sole cue for objectness and reflects scenarios such as a workbench where a person only interacts with a small number of objects at a time.

Empirically, we validate our model against several ablations and baselines. We compare our approach to existing unsupervised multi-object segmentation methods achieving state-of-the-art performance. We demonstrate particularly strong performance in visually complex scenes even with unseen objects and textures at test time. Our experiments in comparison to *image-based* models and, in particular, adding our motion-aware formulation to existing models shows substantial improvements, confirming that motion is an important cue to learn objectness. Furthermore, we show that our learned segmenter, which operates on still images, produces better segmentation results than current *video-based* methods that use motion information at test time. Finally, we also apply our method to real-world self-driving scenarios where we show superior performance to prior work.

## 2 Related Work

**Multi-object decomposition.**  Learning unsupervised object segmentation for static scenes is a well-researched problem in computer vision [5, 9, 11, 12, 13, 14, 17, 18, 23, 34, 37, 45, 46, 56]. These methods aim to decompose the scene into constituent parts, *e.g.* the different foreground objects and the background. Glimpse-based methods [9, 14, 23, 34] find input patches (glimpses) that contain the objects in the scene. These methods learn object descriptors that encode their properties (*e.g.* position, number, size of the objects) using variational inference, composing glimpses into the final picture. More related to ours are approaches that learn per-pixel object masks [5, 11, 12, 13, 37]. MoNet [5] and IODINE [19] employ multiple encoding-decoding steps to sequentially explain the scene as a collection of regions. Slot Attention [37] uses a multi-step soft clustering-like scheme to find the regions simultaneously. In all cases, learning is posed as an image reconstruction problem. In order to align learnable slots with semantic objects, models have to make efficient use of a limited representation available for each region, such as learning to only explain visual appearance. This principle, however, is difficult to extend to visually complex data [27] and relies on custom specialized architectures. Instead, our method allows for any standard segmentation architecture to be used, which we train to predict regions that are most likely described by rigid motion patterns.

**Video-based multi object decomposition.**  Another line of work extends the unsupervised object decomposition problem to videos [21, 24, 26, 29, 30, 31, 33, 44, 52, 54, 55, 59]. Many of these methods work mainly with simpler datasets [30, 44, 59] and require sequential frames for training. For example, SCALOR [24] is a glimpse-based method that discovers and propagates objects across frames to learn intermediate object latents. SIMONe [26] processes the whole video at once, learning both temporal scene representation and time-invariant object representations simultaneously. Slot Attention for Video (SAVi) [28] poses the multi-object problem as optical-flow prediction using

sequential frames as input. The internal slot-attention mechanism drives the network to learn regions that move in a simple and consistent manner. Different to our work, it does not assume a specific motion model but relies on directly regressing the flow. It is computationally more expensive and struggles when only one or few frames are available.

**Unsupervised video object segmentation.** Unsupervised video object segmentation (VOS) is a popular problem in computer vision [10, 15, 32, 38, 43, 49, 51, 57, 58], that focuses on extracting the most salient object in the scene. Many of the approaches treat the problem as a motion segmentation task as the background typically shows a dominant motion independent of the salient object. Motion Grouping [57] employs the Slot Attention architecture to reconstruct optical flow from itself, avoiding appearance information entirely. In [6] objects are iteratively explained away starting from confident flow segments, which are refined using a graph propagation method. Another related line of work [8, 40, 41, 50] employs approximate motion models. These approaches rely on a point estimate of the motion model parameters. In contrast, we adopt a more principled probabilistic approach, placing a prior on the motion parameters and integrating them out. To deal with flow outliers that do not conform to a rigid motion model, Mahendran et al. [40] use a histogram matching-based loss and GWM [8] over-segments the scene relying on spectral clustering to produce a binary segmentation during inference. Instead, we model the noise in our formulation directly. Finally, Meunier et al. [41] rely on flow as input, limiting the method to videos only.

## 3 Method

Let a frame $\mathcal{I} \in \mathbb{R}^{3 \times H \times W}$ of a video and its optical flow $\mathbf{f} \in \mathbb{R}^{2 \times H \times W}$ be defined on the $H \times W$ lattice. The optical flow is a local summary of the motion from one frame to the next. We use it to supervise a network $\Phi$ that, given the (single) image $\mathcal{I}$ as input, predicts soft assignments of each pixel to up to $K$ different image regions, outputting an $H \times W$ collection of probability vectors $\Phi(\mathcal{I}) \in \hat{\Delta}_K^{H \times W} \subset [0,1]^{K \times H \times W}$, where $\hat{\Delta}_K$ is the $K-1$-dimensional simplex. The quality of the regions is measured based on how likely they contain *flow patterns* typical of the motion of independent objects.

In more detail, we represent the predicted image regions by a hard $K$-way pixel assignment (mask) $\mathbf{m} \in \Delta_K^{H \times W} \subset \{0,1\}^{K \times H \times W}$, where $\Delta_K$ is the space of $K$-dimensional one-hot vectors. Each mask is a sample from the categorical distribution output by the network, *i.e.* $\mathbf{m} \sim p_\Phi(\mathbf{m} \mid \mathcal{I}) = \text{Categorical}[\Phi(\mathcal{I})]$. Note that there is one categorical distribution for each pixel and that these are mutually independent.

We then assume that the flow depends only on the regions, in the sense that $p_\Phi(\mathbf{f}, \mathbf{m} \mid \mathcal{I}) = p(\mathbf{f} \mid \mathbf{m}) \, p_\Phi(\mathbf{m} \mid \mathcal{I})$, where $p(\mathbf{f} \mid \mathbf{m})$ is a model of the distribution of the flow field given the regions. The likelihood of the modeled $\Phi$ is bounded by:

$$\log p_\Phi(\mathbf{f} \mid \mathcal{I}) = \log \mathbb{E}_{\mathbf{m} \sim p_\Phi(\mathbf{m} \mid \mathcal{I})} [p(\mathbf{f} \mid \mathbf{m})] \geq \mathbb{E}_{\mathbf{m} \sim p_\Phi(\mathbf{m} \mid \mathcal{I})} [\log p(\mathbf{f} \mid \mathbf{m})].$$

Furthermore, inspired by ELBO, we regularize the model's prediction $p_\Phi(\mathbf{m} \mid \mathcal{I})$ by taking its KL divergence from a uniform prior $p_0(\mathbf{m})$, obtaining the learning objective

$$\mathcal{L}_\beta = \mathbb{E}_{\mathbf{m} \sim p_\Phi(\mathbf{m} \mid \mathcal{I})} [-\log p(\mathbf{f} \mid \mathbf{m})] + \beta D_{\text{KL}} (p_\Phi(\mathbf{m} \mid \mathcal{I}) \,\|\, p_0(\mathbf{m})). \tag{1}$$

Next, we introduce the closed-form motion model $p(\mathbf{f} \mid \mathbf{m})$ in Eq. (1) and then explain how the Gumbel-Softmax trick can be used to train the network.

**Approximate motion models for optical flow.** We now turn to describing the models of motion used in our work, which play a role in assessing the likelihood of optical flow $p(\mathbf{f} \mid \mathbf{m})$. Optical flow measures the coordinate change of pixels between neighboring frames, which arises due to the motion of the camera and objects. We consider rigid-body motion of some object $k$.

Let $\mathbf{x}_k^t, \mathbf{y}_k^t \in \mathbb{R}^{n_k}$ be the spatial locations of the pixels belonging to region/object $k$ at time $t$, where $n_k$ is the number of pixels in the region. For convenience, we stack the coordinates in a single vector $\Omega_k^t = (\mathbf{x}_k^t, \mathbf{y}_k^t) \in \mathbb{R}^{2n_k}$. The pixels comprising this object undergo coordinate change from $\Omega_k^t$ to $\Omega_k^{t+1}$, giving rise to the optical flow for this object as $\mathbf{f}_k = \Omega_k^{t+1} - \Omega_k^t$. We assume this underlying 3D rigid-body motion can be approximated using a linear 2D parametric model $\Pi_\theta$ with parameters $\theta$, so that:

$$\Omega_k^{t+1} = \Pi_\theta(\Omega_k^t) + \epsilon, \qquad \mathbf{f}_k = \Pi_\theta(\Omega_k^t) - \Omega_k^t + \epsilon, \tag{2}$$

where $\epsilon$ captures the residual error of the approximation. Several forms of models are available (see [1, 4] for an overview). Here, we consider two such models: the translation of an object within the camera plane, and an affine motion, given respectively by linear functions:

$$\Pi_\theta^{\text{tr}}(\Omega_k^t) = \Omega_k^t + \underbrace{\begin{bmatrix} \mathbf{1}_{n_k} & 0 \\ 0 & \mathbf{1}_{n_k} \end{bmatrix}}_{P_k^{\text{tr}}} \begin{pmatrix} \theta_1 \\ \theta_2 \end{pmatrix}, \quad \Pi_\theta^{\text{aff}}(\Omega_k^t) = \underbrace{\begin{bmatrix} \mathbf{x}_t & \mathbf{y}_t & \mathbf{1}_{n_k} & 0 & 0 & 0 \\ 0 & 0 & 0 & \mathbf{x}_t & \mathbf{y}_t & \mathbf{1}_{n_k} \end{bmatrix}}_{P_k^{\text{aff}}} \begin{pmatrix} \theta_1 \\ \vdots \\ \theta_6 \end{pmatrix}, \quad (3)$$

where we use $\mathbf{1}_{n_k}$ is a vector of $n_k$ ones and matrix $P_k$ contains the coefficients of the model.

The affine model supports object rotation, scaling and shearing in addition to translation. It is often a sufficient approximation to real-world optical flow, provided the objects are rigid, convex, and mainly rotating in-plane.

We can then use the motion equations (2) to construct the distribution $p(\mathbf{f} \mid \mathbf{m})$ by assuming a prior on the motion parameters and by marginalizing over it. Specifically, denote by $\mathbf{m}_k$ the $k$-th slice of the tensor $\mathbf{m}$ encoding the regions (*i.e.* the mask of the $k$-th region). We assume that regions are statistically independent and decompose the log-likelihood $p(\mathbf{f} \mid \mathbf{m})$ as:

$$\log p(\mathbf{f} \mid \mathbf{m}) = \sum_k \log p(\mathbf{f}_k \mid \mathbf{m}_k) = \sum_k \int \log p(\mathbf{f}_k, \theta_k \mid \mathbf{m}_k) d\theta_k. \quad (4)$$

Assuming that each object has i.i.d. parameters $\theta_k$ with a Gaussian prior $\mathcal{N}(\theta; \mu, \Sigma)$, and assuming $\epsilon$ is a zero-mean noise with variance $\sigma^2$, Eq. (2) gives marginal optical flow likelihood for segment $k$:

$$p(\mathbf{f}_k \mid \mathbf{m}_k) = \mathcal{N}\left(\mathbf{f}_k; \Pi_\mu(\Omega_k) - \Omega_k, P_k \Sigma P_k^\top + \sigma^2 I\right) \quad (5)$$

where $I$ is the identity matrix. A practical issue with Eq. (5) is that, if segment $k$ contains $n_k = \sum_i (\mathbf{m}_k)_i$ pixels, then the covariance matrix $P_k \Sigma P_k^\top + \sigma^2 I$ has dimension $2n_k \times 2n_k$. Inverting such a matrix in the evaluation of the Gaussian log-density is very slow except for very small regions. Furthermore, it is not obvious how to relax Eq. (4) to support gradient-based learning, *e.g.* through the Gumbel-Softmax approximation. We solve these problems in the next section.

**Expressions for the likelihood.** We now derive expressions for Eq. (5) which are efficient and that lead to a natural relaxation for use in the Gumbel-Softmax sampling. Given the definitions $\mathbf{F}_k = \mathbf{f}_k - \Pi_\mu(\Omega_k) + \Omega_k$ and $\Lambda = \Sigma^{-1}$, we can rewrite Eq. (5) as:

$$p(\mathbf{f}_k \mid \mathbf{m}_k) = (2\pi\sigma^2)^{-n_k} \left(\frac{\det S_k}{\det \Lambda}\right)^{-\frac{1}{2}} e^{-\frac{d^2}{2\sigma^2}}, \quad d^2 = \mathbf{F}_k^\top \mathbf{F}_k - \frac{1}{\sigma^2} \mathbf{F}_k^\top P_k S_k^{-1} P_k^\top \mathbf{F}_k, \quad (6)$$

where $S_k = \frac{1}{\sigma^2} P_k^\top P_k + \Lambda$. The significant advantage of this form is that it involves the computation of the inverse and determinant of matrix $S_k$, whose size is only $2 \times 2$ (for the translation model) or $6 \times 6$ (for the affine one), instead of the much larger $2n_k \times 2n_k$.

We can more explicitly introduce the dependency on the region assignments $\mathbf{m}$ by defining selector matrices $R_k \in \{0,1\}^{2n_k \times 2n}$ (with $n = \sum_k n_k = HW$) that extract the $\mathbf{x}$ and $\mathbf{y}$ coordinates of the pixels that belong to the corresponding region, *i.e.* $\Omega_k = R_k \Omega$. We can then also write $\mathbf{F}_k = R_k \mathbf{F}$ and $P_k = R_k P$. Furthermore, the product of the selectors $L_k = R_k^\top R_k \in \{0,1\}^{2n \times 2n}$ can be written directly as a function of the assignment $\mathbf{m}$ as $L_k(\mathbf{m}) = \text{diag}(\mathbf{m}_k, \mathbf{m}_k)$. Plugging these back in Eq. (6), we obtain expressions involving $L_k$ only:

$$n_k = \frac{1}{2}|L_k|_1, \quad S_k = \frac{1}{\sigma^2} P^\top L_k P + \Lambda, \quad d^2 = \mathbf{F}^\top L_k \mathbf{F} - \frac{1}{\sigma^2}(\mathbf{F}^\top L_k P) S_k^{-1}(P^\top L_k \mathbf{F}). \quad (7)$$

**Translation-only model.** Further simplifications are possible for specific models. For instance, for the translation-only model, assuming that $\Lambda = \text{diag}(1/\tau^2, 1/\tau^2)$ then $S_k = \text{diag}(n_k + 1/\tau^2, n_k + 1/\tau^2)$ and, after some calculations, we obtain the expression:

$$-\log p(\mathbf{f} \mid \mathbf{m}) = n \log 2\pi\sigma^2 + \sum_k \log \frac{n_k + \frac{\sigma^2}{\tau^2}}{\frac{\sigma^2}{\tau^2}} + \frac{1}{2\sigma^2} \mathbf{F}^\top \left(I - \sum_k \frac{1}{n_k + \frac{\sigma^2}{\tau^2}} \begin{bmatrix} \mathbf{m}_k \mathbf{m}_k^\top & \\ & \mathbf{m}_k \mathbf{m}_k^\top \end{bmatrix}\right) \mathbf{F}.$$

**Affine model.** For the affine model, the expression for $-\log p(\mathbf{f} \mid \mathbf{m})$ does not simplify as much. Still, by exploiting the structure of matrix $P_k^{\mathrm{aff}}$, we can reduce the calculations to the computation of inverse and determinant of small $3 \times 3$ matrices, which can be implemented efficiently in closed form. Please see the Appendix for the derivation. Unless otherwise stated, the mean vector is set to $\mu = (1\ 0\ 0\ 0\ 1\ 0)^{\top}$ centering the prior on the no-motion point.

**Gumbel-softmax.** In order to train the network using gradient descent, we need a differentiable version of loss (1). To do so, we use the re-parametrizable Gumbel-softmax relaxation [22, 39]. The Gumbel-softmax relaxation replaces categorical samples $\mathbf{m} \in \Delta_K^{H \times W}$ from the distribution $p_\Phi(\mathbf{m} \mid \mathcal{I}) = \mathrm{Categorical}[\Phi(\mathcal{I})]$ with continuous samples $\hat{\mathbf{m}} \in \hat{\Delta}_K^{H \times W}$ from the distribution $\mathrm{GumbelSoftmax}[\Phi(\mathcal{I})]$. We take $N = 3$ samples from this distribution to evaluate the expected negative log-likelihood, further reducing variance. Then we simply replace $\hat{\mathbf{m}}$ for $\mathbf{m}$ in Eq. (1), leading to differentiable quantities.

**Post-processing.** Eq. (5) naturally encourages the model to form larger regions to explain parts of the scene that move in a consistent (under the assumed prior) manner. However, we find that this can also lead to the model grouping together objects that only coincidentally move together (*e.g.* all objects mostly falling due to gravity in one of the datasets). Furthermore, optical flow is ambiguous around object edges and occlusions. To address both the object grouping and occlusion boundary issue, we use a simple post-processing step. We isolate connected components in the model output, selecting the $K$ largest masks, discarding any that are smaller than 0.1% of the image area, and combining the left-over and discarded ones with the largest mask overall.

**Warp loss.** Occasionally, the optical flow used to supervise our model can be noisy as it is estimated by other methods. This noise is also unlikely to be isotropic as some surfaces are easier to estimate that others. Rather than supporting heterogeneous noise and approximation error (Eq. (2)), we instead prioritize parts of the scene covered by higher-quality flow. To this end, we introduce an additional loss term that simply enforces consistency between adjacent frames $\mathcal{I}_1, \mathcal{I}_2$. In particular, it warps the predicted mask distributions $\Phi(\mathcal{I}_1), \Phi(\mathcal{I}_2)$ using the optical flow, weighted by the error of warping the frames themselves, as follows:

$$\mathcal{L}_{\mathrm{warp}}(\mathcal{I}_1, \mathcal{I}_2, f_1, b_2) = w(\mathcal{I}_2, f_1(\mathcal{I}_1)) \cdot d(\Phi(\mathcal{I}_2), f_1(\Phi(\mathcal{I}_1))) \tag{8}$$
$$+ w(\mathcal{I}_1, b_2(\mathcal{I}_2)) \cdot d(\Phi(\mathcal{I}_1), b_2(\Phi(\mathcal{I}_2))),$$
$$w(\mathcal{I}_a, \mathcal{I}_b) = 1 - \mathrm{norm}(|\mathcal{I}_a - \mathcal{I}_b|),$$
$$d(p, q) = D_{\mathrm{KL}}(p \parallel q)/2 + D_{\mathrm{KL}}(q \parallel p)/2,$$

where $f_1(\cdot)$ indicates warping by forward optical flow $f_1$ (or backward $b_2$). The symmetrized KL divergence, $d(\cdot)$, measured agreement between predicted and warped mask distributions, weighted by the absolute error of the warped frames normalized in $[0, 1]$. While the use of this term is not central to our method, we include it to show how tolerance to noisy optical flow can be improved. We do not use the warp loss (Eq. (8)) in our experiments, unless otherwise indicated by (WL). In that case, the final loss is simply sum of the two terms: $\mathcal{L}_\beta + \mathcal{L}_{\mathrm{warp}}$.

## 4 Experiments

Our method lies in between image-based and video-based segmentation approaches, because it uses videos for supervision, but trains an image segmentation network that operates on still images only. We thus evaluate our approach under a number of settings. Firstly, we evaluate how well motion can be used to supervise an object instance segmenter that operates on still images. Secondly, we compare such a segmenter to state-of-the-art object segmentation methods that use motion also at test time (and are thus advantaged compared to our model). We conduct further analysis to validate our modelling assumptions and the model's reliance on the quality of the optical flow used for supervision. Finally, we apply our method to a real-world setting.

### 4.1 Experimental setup

**Datasets.** We evaluate our method on video and still image datasets. For video-based data, we use the Multi-Object Video (MOVi) datasets, released as part of Kubric [20]. Specifically, we employ MOVi-{A,C,D,E} versions. MOVi-A is similar to CLEVR [25] in terms of visual complexity and contains videos of 3–10 falling objects on a simple, gray background. MOVi-C is significantly more

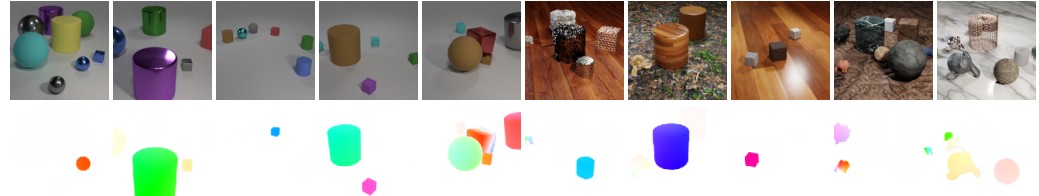

Figure 1: Example images and optical flow of the MOVINGCLEVR and MOVINGCLEVRTEX datasets, which extend CLEVR and CLEVRTEX respectively to short videos based on physics simulation. Note that only a subset of objects is in motion in each frame.

challenging, as it features scanned, textured, common objects on top of backgrounds textured using HDR images. In MOVi-D, the number of objects is increased up to 23. In MOVi-E, the camera is additionaly moving. We use a resolution of $128 \times 128$ and the provided ground truth optical flow.

To evaluate our method on still images, we use CLEVR [25] and CLEVRTEX [27] benchmark suites. Both consist of images depicting 3–10 objects. CLEVR images are simpler with uniformly colored objects with metallic or rubbery materials. CLEVRTEX features more diverse objects with complex textures applied. We also use the OOD and CAMO test sets from CLEVRTEX benchmark. OOD contains out-of-distribution shapes and textures. CAMO has camouflaged objects where the same texture is sampled for objects and the background.

Since our method requires optical flow during training, we extend the implementation of [27] to generate video datasets of CLEVR and CLEVRTEX scenes, where a *subset* of objects contained in each scene are sliding, rolling and colliding based on a physics simulation (Fig. 1). We generate 10k sequences for MOVINGCLEVRTEX and 5k for MOVINGCLEVR, where we retain 1000 and 500 sequences, respectively, for validation. Each sequence is 5 frames long. Dataset details can be found in the Appendix. The evaluation is performed on the original CLEVR and CLEVRTEX test sets.

We also evaluate our method on the real-world KITTI [16] benchmark which depicts street scenes captured from a moving car. We follow the set up of [3], using 147 videos for training and evaluate on the instance segmentation subset which contains 200 annotated validation frames.

**Metrics.** Following prior work [27, 28], we measure performance using two metrics. FG-ARI is the Adjusted Rand Index measured on foreground pixels only (selected using the ground-truth segmentation). Mean Intersection-over-Union (mIoU), is measured through Hungarian matching and averaged across the number of predicted or ground truth components, whichever is higher. When evaluating on videos, we calculate these metrics per-frame.

**Network architecture.** Our method can employ any image segmentation network architecture and train from scratch. Unless otherwise specified, we use Mask2Former [7], using only its semantic segmentation. Following prior work [28, 37], we use a 6-layer CNN backbone on synthetic datasets and ResNet-18 for KITTI. We also experiment with Swin-tiny transformer [36] as the backbone. We use 11 transformer queries which become $K = 11$ slots on CLEVR, CLEVRTEX, and MOVi-A/C. On MOVi-D/E, we set $K = 24$ and $K = 22$ on KITTI. The model takes approximately 48h to train on a single A30 24GB GPU.[2] All training details and hyper-parameters are included in the Appendix.

### 4.2 Unsupervised multi-object segmentation in images

In Table 1, we evaluate our method on the CLEVR [25] and CLEVRTEX [27] benchmarks and compare to prior work. Our method outperforms image models based on appearance reconstruction on both metrics (mIoU and FG-ARI) and across all datasets. The performance gap increases on the visually complex CLEVRTEX, OOD, and CAMO variants, demonstrating the strong inductive bias that motion provides during training, especially when the objects are camouflaged. Note that, in this setting, our model is advantaged compared to the other models in Table 1, as it can observe (through the loss) the optical flow of the training scenes. For this reason, we also train the optical flow-based, unconditional SAVi [28] model. Nevertheless, we find that despite having access to motion information during training, SAVi does not surpass appearance-only models, likely due to only having access to single frames at test time.

---

[2]Approx. total compute in this paper: 100 GPU days for our models, 154 GPU days for comparisons.

Table 1: Benchmark results on CLEVR, CLEVRTEX, CAMO, and OOD comparing FG-ARI and mIoU metrics (see also Appendix for an extended version). Results are a mean of 3 seeds ($\pm\sigma$). Methods above the line are trained on single images, while methods below train on videos. [†] – indicates post-processing.

| | CLEVR | | CLEVRTEX | | OOD | | CAMO | |
|---|---|---|---|---|---|---|---|---|
| Model | FG-ARI↑ | mIoU↑ | FG-ARI↑ | mIoU↑ | FG-ARI↑ | mIoU↑ | FG-ARI↑ | mIoU↑ |
| SPAIR [9] | 77.13± 1.92 | 65.95± 4.02 | 0.00± 0.00 | 0.00±0.00 | 0.00± 0.00 | 0.00±0.00 | 0.00± 0.00 | 0.00±0.00 |
| SPAIR[†] | 77.05± 1.96 | 66.87± 9.65 | 0.00± 0.00 | 0.00±0.00 | 0.00± 0.00 | 0.00±0.00 | 0.00± 0.00 | 0.00±0.00 |
| MN [45] | 72.12± 0.64 | 56.81±0.40 | 38.31± 0.70 | 10.46±0.10 | 37.29± 1.04 | 12.13±0.19 | 31.52± 0.87 | 8.79±0.15 |
| MN[†] | 72.08± 0.62 | 57.61±0.40 | 38.34± 0.73 | 10.34±0.12 | 37.28± 1.07 | 11.97±0.21 | 31.54± 0.87 | 8.77±0.18 |
| MONet [5] | 54.47±11.41 | 30.66±14.87 | 36.66± 0.87 | 19.78±1.02 | 32.97± 1.00 | 19.30±0.37 | 12.44± 0.73 | 10.52±0.38 |
| MONet[†] | 61.36± 7.33 | 45.61± 4.80 | 35.64± 1.17 | 23.59±0.29 | 31.51± 1.46 | 23.04±0.52 | 9.94± 0.50 | 11.31±0.30 |
| SA [37] | 95.89± 2.37 | 36.61±24.83 | 62.40± 2.23 | 22.58±2.07 | 58.45± 1.87 | 20.98±1.59 | 57.54± 1.01 | 19.83±1.41 |
| SA[†] | 94.88± 1.67 | 37.68±26.56 | 61.60± 2.29 | 21.96±1.79 | 57.41± 1.92 | 20.60±1.45 | 56.85± 1.12 | 19.42±1.42 |
| IODINE [19] | 93.81± 0.76 | 45.14±17.85 | 59.52± 2.20 | 29.17±0.75 | 53.20± 2.55 | 26.28±0.85 | 36.31± 2.57 | 17.52±0.75 |
| IODINE[†] | 93.68± 0.83 | 44.20±18.67 | 60.63± 2.50 | 29.40±1.10 | 54.92± 2.24 | 27.96±0.81 | 38.29± 1.40 | 18.87±0.52 |
| DTI-S [42] | 89.54± 1.44 | 48.74± 2.17 | 79.90± 1.37 | 33.79±1.30 | 73.67± 0.98 | 32.55±1.08 | 72.90± 1.89 | 27.54±1.55 |
| DTI-S[†] | 89.86± 1.78 | 53.38± 3.51 | 79.86± 1.36 | 32.20±1.49 | 73.60± 0.97 | 30.74±1.22 | 72.89± 1.88 | 26.30±1.57 |
| GNM [23] | 65.05± 4.19 | 59.92± 3.72 | 53.37± 0.67 | 42.25±0.18 | 48.43± 0.86 | 40.84±0.30 | 15.73± 0.89 | 17.56±0.74 |
| GNM[†] | 65.67± 4.23 | 63.38± 3.76 | 53.38± 0.67 | 44.30±0.19 | 48.44± 0.86 | 42.87±0.28 | 15.72± 0.89 | 18.53±0.75 |
| SAVi [28] | — | — | 49.54 | 31.88 | 42.68 | 30.31 | 42.67 | 29.60 |
| Ours | 91.69± 0.30 | 66.70± 0.32 | 90.80± 0.22 | 55.07±0.44 | 76.01± 0.56 | 46.84±0.20 | 72.78± 1.31 | 42.30±1.09 |
| Ours [†] | **95.94**± 0.43 | **84.86**± 4.06 | **92.61**± 0.22 | **77.67**±0.25 | **78.24**± 0.43 | **55.54**±0.44 | **77.43**± 0.86 | **56.43**±0.80 |

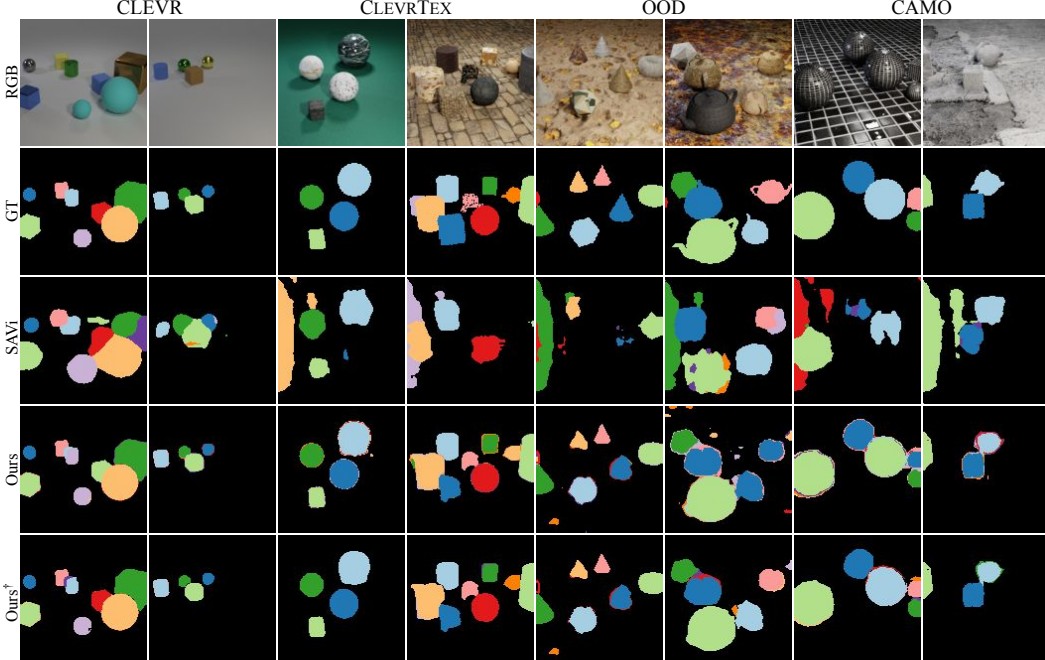

Figure 2: Unsupervised object segmentation on CLEVR and CLEVRTEX benchmarks. Our model is able to segment simple and visually complex scenes. Occasional mistakes around object boundaries and the assignment of different objects to the same component are addressed by post-processing. [†] – indicates post-processing.

Post-processing helps improve results further. As shown in Fig. 2, separating connected components in post-processing distinguishes objects that might be assigned to the same mask and suppresses boundary segments that tend to group difficult occlusion boundaries. For a fairer comparison, we also test if this post-processing improves the results of other methods but only obtain mixed results.

### 4.3 Unsupervised multi-object segmentation in video

We now evaluate our approach on video segmentation, where motion is available at test time. We report the performance of our method in Table 2 compared to video-based models: SCALOR [24],

Table 2: Segmentation results on MOVi datasets. Mean $\pm$ standard error (5 seeds). We calculate metric for each frame. All values in %. (WL) marks use of warp loss. $^{\dagger}$ – indicates post-processing.

| Model | MOVi-A | | MOVi-C | | MOVi-D | | MOVi-E | |
|---|---|---|---|---|---|---|---|---|
| | FG-ARI↑ | mIoU↑ | FG-ARI↑ | mIoU↑ | FG-ARI↑ | mIoU↑ | FG-ARI↑ | mIoU↑ |
| GWM [8] | 70.30 | 42.27 | 49.98 | 30.17 | 39.78 | 18.38 | 42.50 | 18.74 |
| SCALOR [24] | 59.57 | 44.41 | 40.43 | 22.54 | – | – | – | – |
| SAVi [28] | **88.30** | 62.69 | 43.26 | 31.92 | 43.45 | 10.60 | 17.39 | 5.75 |
| Ours | 84.01±0.72 | 60.08±1.47 | 61.18±0.84 | 34.72±0.17 | **55.74**±1.02 | 23.50±0.35 | 62.62±0.92 | 25.78±0.27 |
| Ours$^{\dagger}$ | 85.41±1.00 | **76.19**±2.05 | **61.24**±0.85 | **37.26**±0.33 | 55.18±0.94 | **25.21**±0.29 | **63.11**±0.91 | **28.59**±0.29 |
| Ours$^{\dagger}$ (Swin + WL) | 90.08 | 84.76 | 67.64 | 52.17 | 66.41 | 30.40 | 72.73 | 35.30 |

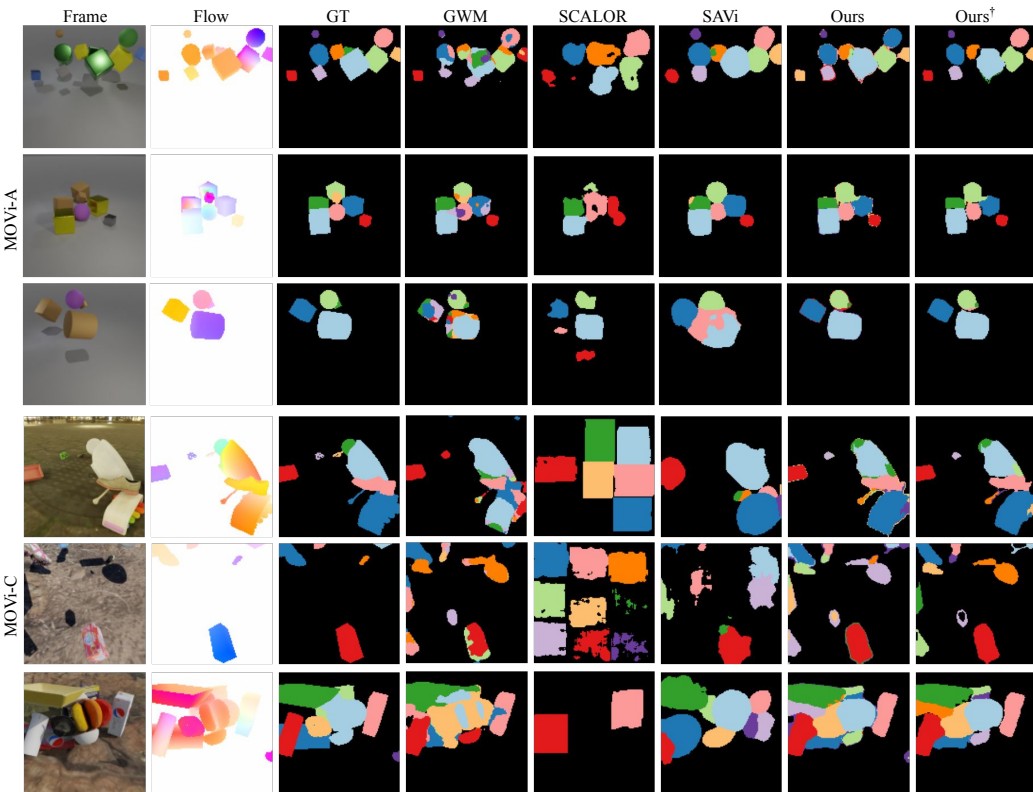

Figure 3: Qualitative comparisons on MOVi-A and MOVi-C. Our method performs consistently well compared to other methods. GWM suffers from oversegmentation where SCALOR has undersegmentation issue. Among the related methods, SCALOR fails to discover all the objects and SAVi's object boundaries are coarser $^{\dagger}$– indicates post-processing.

*unconditional* SAVi [28], and GWM [8]; the latter two also use optical flow supervision. It is important to note that SAVi and SCALOR make predictions jointly over all frames of a video, that allows them to actually see the objects in motion at test time. Despite this comparison being unfair to our approach, which operates on a single frame at a time, we achieve competitive results. On the visually simpler MOVi-A, our method has more than 10% lead over SAVi in mIoU, but performs slightly worse in terms of FG-ARI. On the more complex MoVi-C/D/E datasets our model shows strong performance, outperforming prior work on both metrics, with the performance gap again increasing with data complexity. Finally, we experiment with a version of the model using a deeper backbone (Swin) and the warp loss (Eq. (8)). Though not necessary to achieve state-of-art results, this drastically improves performance on all metrics and dataset versions. In Fig. 3, we also compare these models qualitatively, demonstrating that the different objects are overall better captured by our model with more refined boundaries, which explains the higher mIoU.

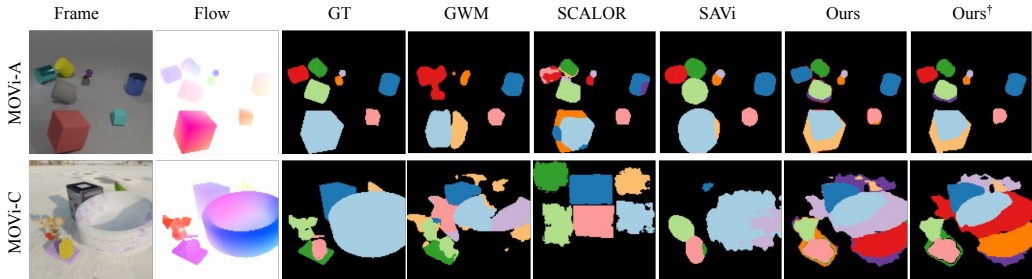

Figure 4: Failure cases on MOVi-A and MOVi-C. Our method has difficulty with objects that typically exhibit complex motion. It results in an over-segmentation of the object. Due to inherently imprecise optical flow near the boundaries our method also has a tendency to segment boundary pixels into a separate mask, which we fix with our post-processing step. $^{\dagger}$– indicates post-processing.

Table 3: Model ablations. (3a) replacing flow supervision with an unsupervised flow method, (3b) compares a translation-only with the affine flow model, and (3c) adds our motion-based formulation to an appearance-only model. All models with post-processing applied.

(a) Choice of Optical Flow Method

| | MOVi-A | | MOVi-C | |
|---|---|---|---|---|
| **Optical Flow** | **FG-ARI↑** | **mIoU↑** | **FG-ARI↑** | **mIoU↑** |
| SMURF [47] | 80.17 | 26.3 | 61.21 | 28.77 |
| Ground Truth | 83.48 | 72.61 | 58.59 | 35.67 |

(b) Choice of Motion Model

| | MOVi-A | | MOVi-C | |
|---|---|---|---|---|
| **Motion Mdl.** | **FG-ARI↑** | **mIoU↑** | **FG-ARI↑** | **mIoU↑** |
| Translation | 66.03 | 59.94 | 39.77 | 32.23 |
| Affine | 83.48 | 72.61 | 58.59 | 35.67 |

(c) Adding motion awareness to appearance-only models

| | | CLEVRTEX | | OOD | | CAMO | |
|---|---|---|---|---|---|---|---|
| **Model** | **Train data** | **FG-ARI↑** | **mIoU↑** | **FG-ARI↑** | **mIoU↑** | **FG-ARI↑** | **mIoU↑** |
| GNM [23] | CLEVRTEX | 53.38 | 44.30 | 48.44 | 42.87 | 15.72 | 18.53 |
| GNM [23] | MOVINGCLEVRTEX | 18.01 | 31.47 | 15.57 | 15.57 | 0.21 | 14.68 |
| GNM+Our Loss | MOVINGCLEVRTEX | 63.84 | 55.26 | 59.01 | 48.65 | 51.00 | 47.63 |
| SA [37] | CLEVRTEX | 62.40 | 22.58 | 58.45 | 20.98 | 57.54 | 19.83 |
| SA [37] | MOVINGCLEVRTEX | 61.84 | 21.44 | 58.24 | 20.67 | 57.30 | 18.82 |
| SA+Our Loss | MOVINGCLEVRTEX | 76.60 | 38.12 | 67.01 | 33.95 | 70.59 | 33.05 |

## 4.4 Model ablations

**Optical flow.** In Table 3a, we replace the ground-truth optical flow used so far in our experiments with the one estimated by SMURF [47]. The additional noise impacts our model's accuracy, as evidenced by the significant drop in mIoU (but comparable FG-ARI scores).

**Motion model.** In Table 3b, we compare our affine motion model to the simpler translation-only one. We observe that the ability to describe complex motion patterns with an affine model improves performance over the translation model, which can only represent translation in the camera plane.

**Motion awareness.** Finally, we investigate the effectiveness of our objective in combination with existing appearance-based methods. The goal of this experiment is to understand the advantage of using motion information during training (if available) and to decouple the effect of our formulation from the choice of architecture. To this end, we employ our objective on top of two models based on appearance reconstruction, GNM [23] and SA [37], with no other modifications. We train GNM and SA with and without our loss on videos (MOVINGCLEVRTEX) and evaluate on the corresponding single-image test sets of the CLEVRTEX suite. In Table 3c, we compare these models respectively to the original methods trained on static images (CLEVRTEX). We find that when trained on video data (MOVINGCLEVRTEX), without our loss, both GNM and SA struggle. We attribute this to the reduced number of scenes in MOVINGCLEVRTEX compared to CLEVRTEX. However, we note that using our loss significantly improves the performance of the appearance methods, suggesting the effectiveness of exploiting motion information through our formulation.

## 4.5 Segmentation on real-world data

We now turn to assessing our model's performance in a real-world setting. We follow the setting of Bao et al. [3] and evaluate on KITTI [16], using RAFT [48] to estimate optical flow and ResNet-18 as the backbone of our model, trained from scratch. We lower the input resolution for our model from $368 \times 1,248$ [3] to $288 \times 960$, which enables us to fit on a single GPU. We evaluate at $96 \times 320$ resolution.

As we show in Table 4, our method outperforms prior work on the challenging real-world setting. In the same table, we also consider our method with an additional warp loss term, which further boosts performance. We also experiment with a transformer-based backbone (Swin) which is also pre-trained using self-supervision. Although, not necessary to show state-of-art result, this significantly improves real-world performance.

Table 4: Real-world segmentation results on KITTI. Baseline results from [3]. Bao et al. [3] and our method use RAFT for optical flow. Models above the line use ResNet-18 backbone. (WL) marks use of warp loss.

| Model | KITTI FG-ARI↑ |
|---|---|
| SA [37] | 13.8 |
| MONet [5] | 14.9 |
| SCALOR [35] | 21.1 |
| S-IODINE [19] | 14.4 |
| MCG [2] | 40.9 |
| Bao et al. [3] | 47.1 |
| **Ours** | 50.8 |
| **Ours** (WL) | **51.9** |
| **Ours** (Swin + WL) | 58.3 |

## 4.6 Limitations

Motion is sometimes insufficient to distinguish different objects, for instance because they do not move or because they move similarly. In principle, this should not matter if sufficient motion diversity is observed in the training data as a whole; in practice, our model occasionally merges different object at test time, which we address partly in post-processing. Further improvements could be obtained by choosing a more informative prior $p_0(\mathbf{m})$ in Eq. (1), to capture other desirable properties of objects, such as compactness and connectedness.

Figure 4 shows some failure cases where the affine motion model struggles to capture strong perspective effects caused by non-smooth depth changes in the object geometry. This could be addressed by modeling the object geometry (depth) and the ensuing complex flow patterns. Alternatively, this could be dealt with using a hierarchical segmentation model that can account for geometric discontinuities and self-occlusions. Hierarchical segmentation would also help with pronounced non-rigid motion (*e.g.* humans dancing or animals running) as motion could be explained at the level of object parts.

## 5 Conclusions

We have presented a method that bridges the gap between image-based and video-based scene decomposition, in that it requires only a single image as input, yet exploits motion cues available in videos during training. In comparison to prior work on image-based multi-object segmentation, our approach shows that motion provides useful objectness cues, especially as the visual complexity of a scene increases. Different from video-based approaches, however, our model operates on still images and does not rely on motion to detect or refine objects, which makes it more generally applicable. Finally, we deviate from the common objective of image or flow *reconstruction* and, instead, model the problem by only predicting regions likely to contain affine flow patterns. This does not require a specialized architecture, thus any segmentation network is suitable for this task. Our approach achieves state-of-the-art performance on multiple image and video benchmarks, in simulated and real-world settings, validating the paradigm of training image models using motion.

**Broader impact.** Our work introduces a principled method for unsupervised multi-object segmentation. The work is mainly evaluated on 3D simulated datasets that do not contain people or personal information. Additionally, we evaluate on KITTI, a real-world self-driving dataset, which occasionally contains images of pedestrians. Consent cannot be obtained in this case, but we follow the KITTI terms of usage. We build on top of open source projects, respecting licenses and release all code, trained models and datasets for research purposes. Currently, the application of this approach is mainly limited to simulated imagery. Although the results on KITTI show promise, the immediate broader impact of our work in real-world scenarios, beyond the research community, is limited.

**Acknowledgements** L. K. is funded by EPSRC Centre for Doctoral Training in Autonomous Intelligent Machines and Systems EP/S024050/1. S. C. is supported by a scholarship sponsored by Facebook. I. L., C. R. and A. V. are supported by European Research Council (ERC) grant 2020-CoG-101001212 UNION. I. L. and C. R. are also funded by EPSRC grant VisualAI EP/T028572/1.

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
