# Unsupervised Multi-object Segmentation by Predicting Probable Motion Patterns Supplementary Material

**Laurynas Karazija\*, Subhabrata Choudhury\*,**
**Iro Laina, Christian Rupprecht, Andrea Vedaldi**

Visual Geometry Group
University of Oxford
Oxford, UK
{laurynas,subha,iro,chrisr,vedaldi}@robots.ox.ac.uk

In this supplementary material, we provide additional details for our loss function (Appendix A), including detailed derivation steps, implementation details, and further discussion of the advantages. Appendix C specifies hyperparameters used and how they were selected. We conclude with additional ablation experiments (Appendix D) and results (Appendix E). Project page and code: https://www.robots.ox.ac.uk/~vgg/research/ppmp.

## A   Loss derivation

The key part of our loss function is the likelihood of the optical flow $p(\mathbf{f} \mid \mathbf{m})$, which serves to evaluate how probable is a region of the optical flow carved out by the predicted masks for K regions. We assume that optical flow within a region depends only on the region itself and that other regions have no influence. Intuitively, this is a reasonable assumption, as in large, the movement/flow of an object does not depend on the background or other objects. Thus, enforcing this assumption encourages regions to correspond to objects.

The assessment of probability of optical flow within the region is based on the assumption that objects should be moving rigidly. We use an approximate parametric motion model (Eq. (2)). The parameters of the motion model $\theta$ abstract away unknown aspects such as scene geometry and camera intrinsics but enable to translate between assumed 3D rigid motion and 2D optical flow.

We assume that motion parameters $\theta_k \sim \mathcal{N}(\theta; \mu, \Sigma)$ come from a multivariate Gaussian prior. This choice enables expressing marginal-likelihood in closed-form.

We model the error of the approximate motion model as zero-mean isotropic Gaussian noise $\epsilon \sim \mathcal{N}(\epsilon; 0, \sigma^2 I)$.

Following the motion model (Eq. (2)), the optical flow $\mathbf{f}_k$ is an *affine combination* of Gaussian random variables. Using this observation, its distribution is

$$p(\mathbf{f}_k \mid \mathbf{m}_k) = \det(2\pi(P_k \Sigma P_k^\top + \sigma^2 I))^{-1/2} \cdot \exp(-\frac{1}{2}\mathbf{F}_k^\top (P_k \Sigma P_k^\top + \sigma^2 I)^{-1}\mathbf{F}_k), \qquad (9)$$

where $\mathbf{F}_k = \mathbf{f}_k - \Pi_\mu(\Omega_k) + \Omega_k$ is the centered flow within the region $k$. This equation can be slightly simplified by considering its two troublesome parts, the determinant and the quadratic form inside

---

\*Authors contributed equally.

36th Conference on Neural Information Processing Systems (NeurIPS 2022).

the exponent. For the determinant, we note the following:

$$\det(2\pi P_k \Sigma P_k^\top + 2\pi\sigma^2 I) = (2\pi\sigma^2)^{2n_k} \det(1/\sigma^2 P_k \Sigma P_k^\top + I)$$

$$= (2\pi\sigma^2)^{2n_k} \det(\Sigma) \det(1/\sigma^2 P_k^\top P_k + \Sigma^{-1}) \qquad (*)$$

$$= \frac{(2\pi\sigma^2)^{2n_k}}{\det(\Lambda)} \det(\underbrace{1/\sigma^2 P_k^\top P_k + \Lambda}_{S_k}),$$

where in the line marked with (*) we apply matrix determinant lemma, and in the last line we substitute covariance $\Sigma^{-1} = \Lambda$ for the precision matrix. Similarly, the quadratic form in the exponent can be expanded

$$\mathbf{F}_k^\top (P_k \Sigma P_k^\top + \sigma^2 I)^{-1} \mathbf{F}_k = 1/\sigma^2 \mathbf{F}_k^\top (1/\sigma^2 P_k \Sigma P_k^\top + I)^{-1} \mathbf{F}_k$$

$$= 1/\sigma^2 \mathbf{F}_k^\top \left(I - 1/\sigma^2 P_k(1/\sigma^2 P_k^\top P_k + \Lambda)^{-1} P_k^\top\right) \mathbf{F}_k \qquad (\dagger)$$

$$= 1/\sigma^2 \mathbf{F}_k^\top \mathbf{F}_k - (1/\sigma^2)^2 \mathbf{F}_k^\top P_k \underbrace{(1/\sigma^2 P_k^\top P_k + \Lambda)^{-1}}_{S_k} P_k^\top \mathbf{F}_k,$$

where in the line marked with (†) the Woodbury identity is applied. The optical flow for the whole image is modeled as a joint of independent flow regions $\mathbf{f}_k$, giving the log-likelihood as

$$\log p(\mathbf{f} \mid \mathbf{m}) = \sum_k \log p(\mathbf{f}_k \mid \mathbf{m}_k)$$

$$= -\frac{1}{2}\left(2\log(2\pi\sigma^2)\sum_k n_k + \sum_k \log\frac{\det S_k}{\det \Lambda} + 1/\sigma^2 \sum_k \mathbf{F}_k^\top \mathbf{F}_k - (1/\sigma^2)^2 \sum_k \mathbf{F}_k^\top P_k S_k^{-1} P_k^\top \mathbf{F}_k\right)$$

$$= -\frac{1}{2}\left(2\log(2\pi\sigma^2)n + \sum_k \log\frac{\det S_k}{\det \Lambda} + 1/\sigma^2\mathbf{F}^\top \mathbf{F} - (1/\sigma^2)^2 \sum_k \mathbf{F}^\top L_k P S_k^{-1} P^\top L_k \mathbf{F}\right), \quad (10)$$

where in the last line we introduced $n = \sum_k n_k = HW$, the number of pixels in the image. We also use product of selector matrices $L_k = R_k^\top R_k = \mathrm{diag}(\mathbf{m}_k, \mathbf{m}_k) = L_k^\top$ such that $\mathbf{F}_k^\top P_k = \mathbf{F}^\top L_k P$. This explicitly includes masks in the expression. Finally, we use the fact that regions partition the full image $\sum_k \mathbf{F}_k^\top \mathbf{F}_k = \sum_k \sum_i (\mathbf{F}_k)_i^2 = \mathbf{F}^\top \mathbf{F}$. We now manipulate Eq. (10) using specific details of the motion model to arrive at expressions that are convenient to implement in code.

## A.1 Implementation details

**Translation-only likelihood**    We assume that translation along x and y directions is independent, such that $\theta$ prior is a zero-mean Gaussian with isotropic covariance $\tau^2 I$. $P_k^{\mathrm{tr}} = \mathrm{diag}(\mathbf{1}_{n_k}, \mathbf{1}_{n_k})$ and by extension $P^{\mathrm{tr}} = \mathrm{diag}(\mathbf{1}_n, \mathbf{1}_n)$. The matrix $S_k$ simplifies to

$$S_k = 1/\sigma^2 P_k^\top P_k + \Lambda = 1/\sigma^2 \begin{pmatrix} \mathbf{1}_{n_k}^\top \mathbf{1}_{n_k} & 0 \\ 0 & \mathbf{1}_{n_k}^\top \mathbf{1}_{n_k} \end{pmatrix} + 1/\tau^2 I = 1/\sigma^2(n_k + \sigma^2/\tau^2)I\,,$$

such that

$$\det S_k = (1/\sigma^2(n_k + \sigma^2/\tau^2))^2, \quad \log\frac{\det S_k}{\det \Lambda} = 2\log\frac{n_k + \sigma^2/\tau^2}{\sigma^2/\tau^2}, \quad S_k^{-1} = (1/\sigma^2(n_k + \sigma^2/\tau^2))^{-1}I.$$

Writing $\mathbf{F}^\top = (\mathbf{u}^\top, \mathbf{v}^\top)$ to denote x and y components of the flow, respectively, the term reduces

$$(1/\sigma^2)^2 \sum_k \mathbf{F}^\top L_k P S_k^{-1} P^\top L_k \mathbf{F} = 1/\sigma^2 \sum_k \mathbf{F}^\top L_k^\top P P^\top L_k \mathbf{F} \frac{1}{n_k + \sigma^2/\tau^2}$$

$$= 1/\sigma^2 \sum_k \frac{1}{n_k + \sigma^2/\tau^2}\begin{bmatrix}\mathbf{u}^\top & \mathbf{v}^\top\end{bmatrix}\begin{bmatrix}\mathbf{m}_k \mathbf{m}_k^\top & \\ & \mathbf{m}_k \mathbf{m}_k^\top\end{bmatrix}\begin{bmatrix}\mathbf{u} \\ \mathbf{v}\end{bmatrix}$$

$$= 1/\sigma^2 \sum_k \frac{1}{n_k + \sigma^2/\tau^2}\left((\mathbf{u}^\top \mathbf{m}_k)^2 + (\mathbf{v}^\top \mathbf{m}_k)^2\right)$$

$$= 1/\sigma^2 \sum_k \frac{n_k^2}{n_k + \sigma^2/\tau^2}(\bar{u}_k^2 + \bar{v}_k^2)\,,$$

where in the last line we introduced mean flow $\bar{u}_k = n_k^{-1}\mathbf{u}^\top \mathbf{m}_k$ and $\bar{v}_k = n_k^{-1}\mathbf{v}^\top \mathbf{m}_k$. This gives the negative log-likelihood as

$$
\begin{aligned}
\log p(\mathbf{f} \mid \mathbf{m}) &= \sum_k \log p(\mathbf{f}_k \mid \mathbf{m}_k) \\
&= -\frac{1}{2}\left( 2\log(2\pi\sigma^2)n + \sum_k \log\frac{\det S_k}{\det \Lambda} + \mathbf{1}/\sigma^2\mathbf{F}^\top\mathbf{F} - (1/\sigma^2)^2 \sum_k \mathbf{F}^\top L_k P S_k^{-1} P^\top L_k \mathbf{F} \right) \\
&= -n\log(2\pi\sigma^2) - \sum_k \log\frac{n_k + \sigma^2/\tau^2}{\sigma^2/\tau^2} - \frac{1}{2\sigma^2}\left( \mathbf{F}^\top\mathbf{F} - \sum_k \frac{n_k^2(\bar{u}_k^2 + \bar{v}_k^2)}{n_k + \sigma^2/\tau^2} \right) \\
&= -n\log(2\pi\sigma^2) - \sum_k \log\frac{n_k + \sigma^2/\tau^2}{\sigma^2/\tau^2} - \frac{1}{2\sigma^2}\sum_{i=1}^n \left( u_i^2 + v_i^2 - \sum_k \frac{n_k(\bar{u}_k^2 + \bar{v}_k^2)}{n_k + \sigma^2/\tau^2}(\mathbf{m}_k)_i \right). \quad (11)
\end{aligned}
$$

In our implementation, we extend this equation further. Writing $w_k = 1 - \sqrt{\frac{\sigma^2/\tau^2}{n_k+\sigma^2/\tau^2}}$, we note that the following sum is equivalent

$$
\begin{aligned}
\sum_{i=1}^n \left(u_i - \sum_k \bar{u}_k w_k (\mathbf{m}_k)_i\right)^2 &= \sum_{i=1}^n \left(u_i^2 - \sum_k 2u_i \bar{u}_k w_k (\mathbf{m}_k)_i + \sum_k (\mathbf{m}_k)_i w_k^2 \bar{u}_k^2\right) \\
&= \sum_{i=1}^n u_i^2 - \sum_k 2\bar{u}_k w_k \sum_{i=1}^n u_i(\mathbf{m}_k)_i + \sum_k w_k^2 \bar{u}_k^2 \sum_{i=1}^n (\mathbf{m}_k)_i \\
&= \sum_{i=1}^n u_i^2 - \sum_k 2\bar{u}_k^2 w_k n_k + \sum_k w_k^2 \bar{u}_k^2 n_k \\
&= \sum_{i=1}^n u_i^2 - \sum_k \bar{u}_k^2 n_k(2w_k - w_k^2) \\
&= \sum_{i=1}^n u_i^2 - \sum_k \bar{u}_k^2 \frac{n_k}{n_k + \sigma^2/\tau^2} \sum_{i=1}^n (\mathbf{m}_k)_i \\
&= \sum_{i=1}^n \left( u_i^2 - \sum_k \frac{n_k \bar{u}_k^2 (\mathbf{m}_k)_i}{n_k + \sigma^2/\tau^2} \right),
\end{aligned}
$$

where in the first line we make use of the fact that masks are one-hot $(\mathbf{m})_i \in \{0,1\}^k$, thus only a single term in the sums over $k$ is non-zero, i.e. $(\sum_k w_k \bar{u}_k(\mathbf{m}_k)_i)^2 = \sum_k w_k^2 \bar{u}_k^2(\mathbf{m}_k)_i$. Using the above insight, the log-likelihood is

$$
\begin{aligned}
\log p(\mathbf{f} \mid \mathbf{m}) = &-n\log(2\pi\sigma^2) - \sum_k \log\frac{n_k + \sigma^2/\tau^2}{\sigma^2/\tau^2} \\
&- \frac{1}{2\sigma^2}\sum_{i=1}^n \left( \left(u_i - \sum_k \bar{u}_k w_k (\mathbf{m}_k)_i\right)^2 + \left(v_i - \sum_k \bar{v}_k w_k (\mathbf{m}_k)_i\right)^2 \right), \quad (12)
\end{aligned}
$$

where

$$
n_k = \sum_{i=1}^n (\mathbf{m}_k)_i, \quad w_k = 1 - \sqrt{\frac{\sigma^2/\tau^2}{n_k+\sigma^2/\tau^2}}.
$$

We then replace $\mathbf{m}$ with $\hat{\mathbf{m}}$ from the Gumbel-Softmax approximation.

**Affine motion likelihood.** For the affine motion model, the full covariance matrix $\Sigma$ prevents significant further simplification. Instead, we transform the log-likelihood (Eq. (10)) to an equivalent form involving only $3 \times 3$ matrices, for which the required determinant and inverse can be calculated

analytically. To that end, we introduce the following auxiliary variables:

$$G_k = \begin{bmatrix} \mathbf{x}_k & \mathbf{y}_k & \mathbf{1}_{n_k} \end{bmatrix}, \quad P_k = \begin{bmatrix} G_k & 0 \\ 0 & G_k \end{bmatrix}, \quad \Sigma^{-1} = \Lambda = \begin{bmatrix} \alpha & \beta \\ \gamma & \delta \end{bmatrix}.$$

Then $S_k$ is

$$S_k = \frac{1}{\sigma^2} P_k^\top P_k + \Lambda = \begin{bmatrix} \frac{1}{\sigma^2} G_k^\top G_k + \alpha & \beta \\ \gamma & \frac{1}{\sigma^2} G_k^\top G_k + \delta \end{bmatrix},$$

with

$$G_k^\top G_k = \begin{pmatrix} \mathbf{x}_k^\top \mathbf{x}_k & \mathbf{x}_k^\top \mathbf{y}_k & \mathbf{x}_k^\top \mathbf{1}_{n_k} \\ \mathbf{x}_k^\top \mathbf{y}_k & \mathbf{y}_k^\top \mathbf{y}_k & \mathbf{y}_k^\top \mathbf{1}_{n_k} \\ \mathbf{x}_k^\top \mathbf{1}_{n_k} & \mathbf{y}_k^\top \mathbf{1}_{n_k} & \mathbf{1}_{n_k}^\top \mathbf{1}_{n_k} \end{pmatrix}.$$

Using this, the determinant is

$$\det S_k = \det(\tfrac{1}{\sigma^2} G_k^\top G_k + \alpha - \beta(\tfrac{1}{\sigma^2} G_k^\top G_k + \delta)^{-1}\gamma) \det(\tfrac{1}{\sigma^2} G_k^\top G_k + \delta).$$

Similarly, the inverse is then

$$S_k^{-1} = \begin{bmatrix} A_k & B_k \\ C_k & D_k \end{bmatrix}, \quad \text{where}$$

$$D_k = (\tfrac{1}{\sigma^2} G_k^\top G_k + \delta - \gamma(\tfrac{1}{\sigma^2} G_k^\top G_k + \alpha)^{-1}\beta)$$

$$C_k = -D_k \gamma(\tfrac{1}{\sigma^2} G_k^\top G_k + \alpha)^{-1}$$

$$B_k = -(\tfrac{1}{\sigma^2} G_k^\top G_k + \alpha)^{-1}\beta D_k$$

$$A_k = (\tfrac{1}{\sigma^2} G_k^\top G_k + \alpha)^{-1} - B_k \gamma(\tfrac{1}{\sigma^2} G_k^\top G_k + \alpha)^{-1}, \text{ such that}$$

$$\mathbf{h}_k = \begin{pmatrix} \mathbf{u}_k^\top \mathbf{x}_k & \mathbf{u}_k^\top \mathbf{y}_k & \mathbf{u}_k^\top \mathbf{1}_{n_k} \end{pmatrix}$$

$$\mathbf{r}_k = \begin{pmatrix} \mathbf{v}_k^\top \mathbf{x}_k & \mathbf{v}_k^\top \mathbf{y}_k & \mathbf{v}_k^\top \mathbf{1}_{n_k} \end{pmatrix}$$

$$\mathbf{F}_k^\top P_k S_k^{-1} P_k^\top \mathbf{F}_k = \mathbf{h}_k A_k \mathbf{r}_k^\top + \mathbf{r}_k C_k \mathbf{h}_k^\top + \mathbf{h}_k B_k \mathbf{r}_k^\top + \mathbf{r}_k D_k \mathbf{r}_k^\top.$$

We implement inner products under Gumbel-Softmax as $\mathbf{a}_k^\top \mathbf{b}_k = \sum_{i=1}^n (\mathbf{a})_i (\mathbf{b})_i (\hat{\mathbf{m}}_k)_i$, for some vectors $\mathbf{a}, \mathbf{b}$. The coordinate vectors $\mathbf{x}_k, \mathbf{y}_k$ have the origin set to the centroid of the predicted region $\begin{pmatrix} x_{c,k} \\ y_{c,k} \end{pmatrix} = n_k^{-1} \begin{pmatrix} \mathbf{x}^\top \mathbf{m}_k \\ \mathbf{y}^\top \mathbf{m}_k \end{pmatrix}$. The expressions can then be substituted back to Eq. (10). We show implementation in Algorithm 1.

## A.2 Further justification

We consider whether the inclusion of the prior on the motion parameters offers any benefits. Consider a simple translation-only model. Since objects are only translating, each pixel in a region should be very close to the mean translation of that region. We can assess the mean for a region as $\begin{bmatrix} \bar{u}_k \mathbf{m}_k \\ \bar{v}_k \mathbf{m}_k \end{bmatrix}$, considering some variance $\sigma^2$ around it:

$$\log \hat{p}(\mathbf{f} \mid \mathbf{m}) = \log \mathcal{N}\left( \begin{bmatrix} \mathbf{u} \\ \mathbf{v} \end{bmatrix}; \sum_k \begin{bmatrix} \bar{u}_k \mathbf{m}_k \\ \bar{v}_k \mathbf{m}_k \end{bmatrix}, \sigma^2 I \right)$$

$$= -n \log(2\pi\sigma^2) - \frac{1}{2\sigma^2} \sum_{i=1}^n \left( (u_i - \sum_k \bar{u}_k(\mathbf{m}_k)_i)^2 + (v_i - \sum_k \bar{v}_k(\mathbf{m}_k)_i)^2 \right).$$

Such model, up to a scaling factor, was already considered for features [? ] and optical flow [2]. This expression for $\log \hat{p}(\mathbf{f} \mid \mathbf{m})$ can be compared with our version of translation only model Eq. (12). By considering the prior on the motion parameters, we introduce a weighing factor $w_k$ for each mean $\bar{u}_k \mathbf{m}_k$, which discounts the contribution of smaller segments. Similarly, the term $\sum_k \log \frac{n_k + \sigma^2/\tau^2}{\sigma^2/\tau^2} \approx \sum_k \log n_k$ encourages larger masks, since $\sum_k n_k = n$. The prior helps to encode that larger regions should be preferred.

**Algorithm 1** Implementation of negative flow likelihood $-\log p(\mathbf{f} \mid \mathbf{m})$ under affine motion prior. Key quantities in the inner loop are underlined.

---

1: **procedure** NLL(mean $\mu$, covariance $\Sigma$, variance $\sigma^2$, flow $\mathbf{f}$, masks $\mathbf{m}_k$, height $H$, width $W$)
2: $\quad (\mu_1 \ \mu_2 \ \mu_3 \ \mu_4 \ \mu_5 \ \mu_6) \leftarrow \mu$
3: $\quad \mathbf{x}, \mathbf{y} \leftarrow \mathrm{lattice}(H, W)$
4: $\quad \begin{bmatrix} \mathbf{u} \\ \mathbf{v} \end{bmatrix} \leftarrow \mathbf{f}$
5: $\quad \Lambda \leftarrow \Sigma^{-1}$
6: $\quad \begin{bmatrix} \alpha & \beta \\ \gamma & \delta \end{bmatrix} \leftarrow \Lambda$
7: $\quad$ **for all** $k$ regions **do**
8: $\qquad n_k \leftarrow \sum_i (\mathbf{m}_k)_i$
9: $\qquad \hat{\mathbf{x}}_k \leftarrow \mathbf{x} - n_k^{-1} \mathbf{x}^\top \mathbf{m}_k$ $\qquad\qquad\qquad\qquad\qquad\qquad$ ▷ Set origin to centroid
10: $\qquad \hat{\mathbf{y}}_k \leftarrow \mathbf{y} - n_k^{-1} \mathbf{y}^\top \mathbf{m}_k$
11: $\qquad \mathbf{x}_k^\top \mathbf{x}_k \leftarrow \sum_i (\hat{\mathbf{x}}_k)_i^2 (\mathbf{m}_k)_i$
12: $\qquad \mathbf{x}_k^\top \mathbf{y}_k \leftarrow \sum_i (\hat{\mathbf{x}}_k)_i (\hat{\mathbf{y}}_k)_i (\mathbf{m}_k)_i$
13: $\qquad \mathbf{y}_k^\top \mathbf{y}_k \leftarrow \sum_i (\hat{\mathbf{y}}_k)_i^2 (\mathbf{m}_k)_i$
14: $\qquad \mathbf{x}_k^\top \mathbf{1}_{n_k} \leftarrow \sum_i (\hat{\mathbf{x}}_k)_i (\mathbf{m}_k)_i$
15: $\qquad \mathbf{y}_k^\top \mathbf{1}_{n_k} \leftarrow \sum_i (\hat{\mathbf{y}}_k)_i (\mathbf{m}_k)_i$
16: $\qquad G_k^\top G_k \leftarrow \begin{pmatrix} \mathbf{x}_k^\top \mathbf{x}_k & \mathbf{x}_k^\top \mathbf{y}_k & \mathbf{x}_k^\top \mathbf{1}_{n_k} \\ \mathbf{x}_k^\top \mathbf{y}_k & \mathbf{y}_k^\top \mathbf{y}_k & \mathbf{y}_k^\top \mathbf{1}_{n_k} \\ \mathbf{x}_k^\top \mathbf{1}_{n_k} & \mathbf{y}_k^\top \mathbf{1}_{n_k} & n_k \end{pmatrix}$
17: $\qquad D_k \leftarrow (1/\sigma^2 G_k^\top G_k + \delta - \gamma (1/\sigma^2 G_k^\top G_k + \alpha)^{-1} \beta)$
18: $\qquad C_k \leftarrow -D_k \gamma (1/\sigma^2 G_k^\top G_k + \alpha)^{-1}$
19: $\qquad B_k \leftarrow -(1/\sigma^2 G_k^\top G_k + \alpha)^{-1} \beta D_k$
20: $\qquad A_k \leftarrow (1/\sigma^2 G_k^\top G_k + \alpha)^{-1} - B_k \gamma (1/\sigma^2 G_k^\top G_k + \alpha)^{-1}$
21: $\qquad \mathbf{u}_k^\top \mathbf{x}_k \leftarrow \sum_i ((\mathbf{u})_i)((\mathbf{x})_i - x_{c,k})(\mathbf{m}_k)_i$
22: $\qquad \hat{\mathbf{u}}_k \leftarrow \mathbf{u} - ((\mu_1 - 1)\hat{\mathbf{x}} + \mu_2 \hat{\mathbf{y}} + \mu_3)$ $\qquad\qquad$ ▷ Center flow according to mean motion
23: $\qquad \hat{\mathbf{v}}_k \leftarrow \mathbf{v} - ((\mu_5 - 1)\hat{\mathbf{y}} + \mu_4 \hat{\mathbf{x}} + \mu_6)$
24: $\qquad \underline{\mathbf{F}_k^\top \mathbf{F}_k} \leftarrow \sum_i (\hat{\mathbf{u}}_k)_i^2 (\mathbf{m}_k)_i + \sum_i (\hat{\mathbf{v}}_k)_i^2 (\mathbf{m}_k)_i$
25: $\qquad \underline{\mathbf{u}_k^\top \mathbf{x}_k} \leftarrow \sum_i (\hat{\mathbf{u}}_k)_i (\hat{\mathbf{x}}_k)_i (\mathbf{m}_k)_i$
26: $\qquad \mathbf{u}_k^\top \mathbf{y}_k \leftarrow \sum_i (\hat{\mathbf{u}}_k)_i (\hat{\mathbf{y}}_k)_i (\mathbf{m}_k)_i$
27: $\qquad \mathbf{u}_k^\top \mathbf{1}_{n_k} \leftarrow \sum_i (\hat{\mathbf{u}}_k)_i (\mathbf{m}_k)_i$
28: $\qquad \mathbf{h}_k \leftarrow \begin{pmatrix} \mathbf{u}_k^\top \mathbf{x}_k & \mathbf{u}_k^\top \mathbf{y}_k & \mathbf{u}_k^\top \mathbf{1}_{n_k} \end{pmatrix}$
29: $\qquad \mathbf{v}_k^\top \mathbf{x}_k \leftarrow \sum_i (\hat{\mathbf{v}}_k)_i (\hat{\mathbf{x}}_k)_i (\mathbf{m}_k)_i$
30: $\qquad \mathbf{v}_k^\top \mathbf{y}_k \leftarrow \sum_i (\hat{\mathbf{v}}_k)_i (\hat{\mathbf{y}}_k)_i (\mathbf{m}_k)_i$
31: $\qquad \mathbf{v}_k^\top \mathbf{1}_{n_k} \leftarrow \sum_i (\hat{\mathbf{v}}_k)_i (\mathbf{m}_k)_i$
32: $\qquad \mathbf{r}_k \leftarrow \begin{pmatrix} \mathbf{v}_k^\top \mathbf{x}_k & \mathbf{v}_k^\top \mathbf{y}_k & \mathbf{v}_k^\top \mathbf{1}_{n_k} \end{pmatrix}$
33: $\qquad \underline{\mathbf{F}_k^\top P_k S_k^{-1} P_k^\top \mathbf{F}_k} \leftarrow \mathbf{h}_k A_k \mathbf{r}_k^\top + \mathbf{r}_k C_k \mathbf{h}_k^\top + \mathbf{h}_k B_k \mathbf{r}_k^\top + \mathbf{r}_k D_k \mathbf{r}_k^\top$
34: $\qquad \underline{\det S_k} \leftarrow \det(1/\sigma^2 G_k^\top G_k + \alpha - \beta(1/\sigma^2 G_k^\top G_k + \delta)^{-1} \gamma) \det(1/\sigma^2 G_k^\top G_k + \delta)$
35: $\quad$ **end for**
36: $\quad$ **return** $HW \log(2\pi\sigma^2) + \frac{1}{2} \sum_k \log \frac{\det S_k}{\det \Lambda} + \frac{1}{2\sigma^2} \sum_k (\mathbf{F}_k^\top \mathbf{F}_k - \mathbf{F}_k^\top P_k S_k^{-1} P_k^\top \mathbf{F}_k)$
37: **end procedure**

---

# B MOVINGCLEVRTEX and MOVINGCLEVR

We extend the implementation of [9] to generate video datasets of CLEVR and CLEVRTEX scenes. We follow original sampling set up of CLEVRTEX – each scene contains a random arrangement of 3–10 objects. We uniformly choose between scenarios where a single random objects is given initial motion, two random objects are provided initial motion or all objects are moving. We sample a random initial translation in XY plane for the object between keyframes 0 and 3, which builds momentum. Physics simulation takes over from keyframe 4. Mass of each object is set to equal its scale (numerically), and we use value of 0.1 for the 'bounciness' parameters. This results in objects sliding, rotating due to shape and friction, and colliding. Collisions can make other objects move. Each simulation is 5 frames long and we render keyframes 4 to 8.

We sample 10000 scenes for MOVINGCLEVRTEX, which gives the same number of frames as the original CLEVRTEX (where each scene had only single frame). For MOVINGCLEVR, we sample 5000 scenes. 1000 and 500 scenes are kept as validation for MOVINGCLEVRTEX and for MOVINGCLEVR, respectively. We use the same rendering and lighting parameters as in [9], except we slightly reduce the scale of the surface displacement mapping for the background. This reduces the visibility of clipping of the detailed object geometry with the background geometry, which might occur due to physics simulation working on simplified meshes.

# C Hyperparameters

Our method can use any segmentation network $\Phi$. We employ the recent Mask2Former[2] architecture, using 6-layer CNN backbone from [10, 12] for simulated datasets and ResNet-18 for KITTI in the main experiments. We also experiment with Swin-tiny transformer as the backbone, as it offers balanced performance in both visually simple and complex settings. We ablate these choices in Appendix D.

The networks are trained with AdamW [? ], with a learning rate of $3 \times 10^{-6}$ and batch size of 32, for 250k iterations. We employ gradient clipping when the 2-norm exceeds 0.01 and linear learning rate warm-up for the first 5k iterations to stabilize the training. The learning rate is reduced by a factor of 10 after 200k iterations. When training with warping loss on MOVi datasets, we found it beneficial to train for longer, for 500k iterations, reducing the learning rate by factor of 10 also after 400k iterations.

We found it beneficial to linearly anneal $\beta$ from 0.1 to -0.1 over 5k iterations, encouraging the network to explore initially but focus on low-entropy distributions in the end. We found this had the effect of encouraging the network to assign background pixels to a single slot.

For the prior, we set $\sigma^2$ (Eq. (10)) to $0.5$ and use $\mu = (1\,0\,0\,0\,1\,0)^\top$. We use the following covariance matrix

$$\Sigma = \begin{pmatrix} 0.005 & 0 & 0 & 0 & 0 & 0 \\ 0 & 0.05 & 0 & 0 & 0 & 0 \\ 0 & 0 & 15 & 0 & 0 & 0 \\ 0 & 0 & 0 & 0.05 & 0 & 0 \\ 0 & 0 & 0 & 0 & 0.005 & 0 \\ 0 & 0 & 0 & 0 & 0 & 15 \end{pmatrix}.$$

## C.1 Settings in ablations

When experimenting with translation-only model, we use the same parameters and settings where possible. We set $\tau^2$, so $0.5, 16.0, 34.0$ for CLEVR/CLEVRTEX, MOVi-A, and MOVi-C, respectively (values picked from specialized covariance matrices, see Appendix D).

For GNM [7], we use implementation and parameters described in [9]. Owning to our loss being lower bound on log-probability, we simply add our loss to existing ELBO loss with hyperparameters above, only changing $\sigma^2 = 0.1$. We hypothesize that lower noise model is beneficial, as it provides stronger learning signal in the early stages when reconstruction is noisy due to untrained VAEs. The

---

[2]Code available from https://github.com/facebookresearch/Mask2Former.

Table 1: Supplementary ablations for our methods. We show the impact segmentation networks have for different datasets Table 1a. We further study the impact of our tuned covariance matrix in Table 1b. Results with post-processing applied.

(a) Choice of Model Architecture

| Architecture | # Params | MOVi-A | | MOVi-C | | MovingClevrTex | |
|---|---|---|---|---|---|---|---|
| | | FG-ARI↑ | mIoU↑ | FG-ARI↑ | mIoU↑ | FG-ARI↑ | mIoU↑ |
| M2F (Swin-tiny) | 47M | 83.48 | 72.61 | 58.59 | 35.67 | 88.80 | 69.62 |
| M2F (ResNet50) | 44M | 83.44 | 68.06 | 60.32 | 34.80 | 90.40 | 67.07 |
| M2F (ResNet18) | 31M | 84.04 | 67.48 | 60.84 | 35.69 | 90.31 | 67.33 |
| MF (Swin-tiny) | 44M | 81.78 | 71.28 | 54.45 | 33.67 | 71.07 | 51.06 |
| U-Net | 31M | 90.79 | 82.85 | 60.28 | 26.62 | 87.03 | 39.66 |

(b) Choice of Covariance Marix

| $\Sigma$ | MOVi-A | | MOVi-C | |
|---|---|---|---|---|
| | FG-ARI↑ | mIoU↑ | FG-ARI↑ | mIoU↑ |
| Generic | 82.32 | 71.70 | 58.12 | 35.79 |
| Tuned | 83.48 | 72.61 | 58.59 | 35.67 |

overfitting to errors of the motion approximation is handled, instead, by the networks balancing between appearance reconstruction and motion explanation objectives.

For SA [12], we also use implementation of [9]. SA loss is multiplied by 100 before adding our formulation.

## C.2 Settings in KITTI

For experiments on KITTI, we replace the backbone to ResNet-18 to match prior work. We reduce batch size to 8, increase learning rate to $10^{-4}$. Learning rate is linearly warmed up for 10 iteration, and reduced by a factor of 10 after 5500 iterations. We employ backbone learning rate multiplier of 0.1. All other settings as before.

## C.3 Model parameters of comparisons

Here we describe implementation and hyperparameters used for comparative methods in our experiments.

**SAVi [10].** We follow the parameters given for *conditional* SAVi-S in their code repository[3], except to make SAVi unconditional we unset the conditioning key and make the slots to be learnable parameters.

**SCALOR [8].** We use the optimized SCALOR parameters mentioned by SAVi [10] to train SCALOR. Particularly we use the MOVI dataset parameters to train MOVi-A and MOVI++ parameters to train MOVi-C.

**GWM [2].** For GWM, we follow the parameters mentioned in the paper, except we do not employ spectral clustering and match the number of components to our settings for each dataset.

## D  Additional ablations

**Segmentation network.** We study the effect of segmentation network in Table 1a. We find that using a much simpler U-Net [? ] architecture is beneficial on MOVi-A which contains visually plain scenes. The U-Net architecture, however, results in performance degradation on visually complex data.

---

[3]Code available from https://github.com/google-research/slot-attention-video.

We also consider a version of Mask2Former architecture that uses much deeper backbone, using Resnet-50 and Resnet-18, instead. We find that the deeper backbones leads to similar performance, indicating that our formulation is not architecture-specific. Finally, we change to MaskFormer architecture, matching the network architecture used in [2], and use smaller Swin-tiny backbone. We find that our loss formulation leads to improved performance still.

**Covariance matrix.** We investigate whether further improvements are possible using specialised versions of the prior. To that end, we offset the mean translation prior to account for the dominantly downward motion of objects on MOVi-A/MOVi-C. We use $\mu_{\mathrm{MOVi-A}} = (1\ 0\ 0\ 0\ 1\ 1.5)^\top$ and $\mu_{\mathrm{MOVi-C}} = (1\ 0\ 0\ 0\ 1\ 1.8)^\top$, respectively.[4] We use the following specialized covariance matrices:

$$
\Sigma_{\mathrm{MOVi-A}} = \begin{pmatrix}
0.006 & -0.00004 & 0 & 0.00004 & 0.001 & 0 \\
-0.00004 & 0.04 & 0 & -0.01 & -0.00008 & 0 \\
0 & 0 & 16 & 0 & 0 & 0 \\
0.00004 & -0.01 & 0 & 0.04 & 0.00004 & 0 \\
0.001 & -0.00008 & 0 & 0.00004 & 0.006 & 0 \\
0 & 0 & 0 & 0 & 0 & 14
\end{pmatrix}
$$

$$
\Sigma_{\mathrm{MOVi-C}} = \begin{pmatrix}
0.02 & 0.00002 & 0 & 0.000009 & 0.002 & 0 \\
0.00002 & 0.03 & 0 & -0.009 & -0.000006 & 0 \\
0 & 0 & 36 & 0 & 0 & 0 \\
0.000009 & -0.009 & 0 & 0.04 & -0.00007 & 0 \\
0.002 & -0.000006 & 0 & -0.00007 & 0.02 & 0 \\
0 & 0 & 0 & 0 & 0 & 34
\end{pmatrix}
$$

We obtain the dataset specific covariance matrices by forming initial estimates using a method described below using only optical flow. We then overwrite the entries to encode our belief that translation should be independent. Finally, we further tuned the values through by taking one search step for MOVi-A and MOVi-C each. In our experiments, we found that increasing diagonal elements (variances) and decreasing off-diagonal elements produced slightly better results.

**Initial covariance estimation.** We form initial estimate for the dataset-specific covariance matrices used in ablation only to start hyperparameter search in a sensible range. This method relies on observation that to form an estimate (1) all objects from a frame are not required – only some are sufficient. Furthermore, for the selected object candidates, (2) precise boundaries are not necessary. The method is as follows:

1. We extract discontinuities from the flow using Sobel filtering and treat these as flow edges.

2. We then only consider regions where the optical flow is larger than zero, identifying foreground.

3. We subtract edge pixels from the candidate foreground mask. This attempts to disconnect any overlapping objects using discontinuity in optical flow.

4. We run connected components algorithm to identify candidate object regions.

5. Within each region, using the motion model (Eq. (2)) we estimate $\hat{\theta}$ by forming least-squares solution. We only considered estimates from regions larger than 100 pixels (for numerical stability) and where the residual error was within the 90% percentile.

6. Initial covariance estimate $\hat{\Sigma}$ is formed by calculating sample covariance over the combined set of inliers and extra $n$ no-motion values to account for stationary regions.

We apply this method on a subset of the data. $n$ is the size of the subset.

We find using the specialized settings above give slight improvement on most metrics (Table 1b), indicating that using more appropriate prior for the data further improves results.

Table 2: Expanded benchmark results on CLEVR, CLEVRTEX, CAMO, and OOD comparing FG-ARI and mIoU metrics. Results are a mean of 3 seed ($\pm\sigma$). Methods above the line are trained on single images, while methods below train on videos.[†] – indicates post-processing.

| Model | CLEVR FG-ARI↑ | CLEVR mIoU↑ | CLEVRTEX FG-ARI↑ | CLEVRTEX mIoU↑ | OOD FG-ARI↑ | OOD mIoU↑ | CAMO FG-ARI↑ | CAMO mIoU↑ |
|---|---|---|---|---|---|---|---|---|
| SPAIR [3] | 77.13± 1.92 | 65.95± 4.02 | 0.00± 0.00 | 0.00±0.00 | 0.00± 0.00 | 0.00±0.00 | 0.00± 0.00 | 0.00±0.00 |
| SPAIR[†] | 77.05± 1.96 | 66.87± 9.65 | 0.00± 0.00 | 0.00±0.00 | 0.00± 0.00 | 0.00±0.00 | 0.00± 0.00 | 0.00±0.00 |
| SPACE [11] | 22.75±14.04 | 26.31±12.93 | 17.53± 4.13 | 9.14±3.46 | 12.71± 3.44 | 6.87±3.32 | 10.55± 2.09 | 8.67±3.50 |
| SPACE[†] | 22.74±14.03 | 27.00±13.69 | 17.52± 4.12 | 9.68±4.10 | 12.71± 3.44 | 7.20±3.75 | 10.54± 2.08 | 9.25±3.95 |
| GenV2 [5] | 57.90±20.38 | 9.48± 0.55 | 31.19±12.41 | 7.93±1.53 | 29.04±11.23 | 8.74±1.64 | 29.60±12.84 | 7.49±1.67 |
| GenV2[†] | 57.78±21.12 | 10.76± 1.39 | 30.55±14.27 | 9.04±0.63 | 28.41±13.20 | 9.96±0.70 | 29.19±14.55 | 8.40±1.00 |
| MN [14] | 72.12± 0.64 | 56.81±0.40 | 38.31± 0.70 | 10.46±0.10 | 37.29± 1.04 | 12.13±0.19 | 31.52± 0.87 | 8.79±0.15 |
| MN[†] | 72.08± 0.62 | 57.61±0.40 | 38.34± 0.73 | 10.34±0.12 | 37.28± 1.07 | 11.97±0.21 | 31.54± 0.87 | 8.77±0.18 |
| MONet [1] | 54.47±11.41 | 30.66±14.87 | 36.66± 0.87 | 19.78±1.02 | 32.97± 1.00 | 19.30±0.37 | 12.44± 0.73 | 10.52±0.38 |
| MONet[†] | 61.36± 7.33 | 45.61± 4.80 | 35.64± 1.17 | 23.59±0.29 | 31.51± 1.46 | 23.04±0.52 | 9.94± 0.50 | 11.31±0.30 |
| SA [12] | 95.89± 2.37 | 36.61±24.83 | 62.40± 2.23 | 22.58±2.07 | 58.45± 1.87 | 20.98±1.59 | 57.54± 1.01 | 19.83±1.41 |
| SA[†] | 94.88± 1.67 | 37.68±26.56 | 61.60± 2.29 | 21.96±1.79 | 57.41± 1.92 | 20.60±1.45 | 56.85± 1.12 | 19.42±1.42 |
| IODINE [6] | 93.81± 0.76 | 45.14±17.85 | 59.52± 2.20 | 29.17±0.75 | 53.20± 2.55 | 26.28±0.85 | 36.31± 2.57 | 17.52±0.75 |
| IODINE[†] | 93.68± 0.83 | 44.20±18.67 | 60.63± 2.50 | 29.40±1.10 | 54.92± 2.24 | 27.96±0.81 | 38.29± 1.40 | 18.87±0.52 |
| eMORL [4] | 93.25± 3.24 | 50.19±22.56 | 55.62± 2.12 | 30.17±2.60 | 49.21± 2.69 | 25.03±1.99 | 37.66± 8.41 | 19.13±4.88 |
| eMORL[†] | 93.09± 2.68 | 49.28±24.28 | 58.59± 1.96 | 31.64±2.22 | 51.97± 2.44 | 26.91±1.69 | 43.83± 7.34 | 22.40±4.35 |
| DTI-S [13] | 89.54± 1.44 | 48.74± 2.17 | 79.90± 1.37 | 33.79±1.30 | 73.67± 0.98 | 32.55±1.08 | 72.90± 1.89 | 27.54±1.55 |
| DTI-S[†] | 89.86± 1.78 | 53.38± 3.51 | 79.86± 1.36 | 32.20±1.49 | 73.60± 0.97 | 30.74±1.22 | 72.89± 1.88 | 26.30±1.57 |
| GNM [7] | 65.05± 4.19 | 59.92± 3.72 | 53.37± 0.67 | 42.25±0.18 | 48.43± 0.86 | 40.84±0.30 | 15.73± 0.89 | 17.56±0.74 |
| GNM[†] | 65.67± 4.23 | 63.38± 3.76 | 53.38± 0.67 | 44.30±0.19 | 48.44± 0.86 | 42.87±0.28 | 15.72± 0.89 | 18.53±0.75 |
| SAVi [10] | — | — | 49.54 | 31.88 | 42.68 | 30.31 | 42.67 | 29.60 |
| Ours | 91.69± 0.30 | 66.70± 0.32 | 90.80± 0.22 | 55.07±0.44 | 76.01± 0.56 | 46.84±0.20 | 72.78± 1.31 | 42.30±1.09 |
| Ours [†] | **95.94**± 0.43 | **84.86**± 4.06 | **92.61**± 0.22 | **77.67**±0.25 | **78.24**± 0.43 | **55.54**±0.44 | **77.43**± 0.86 | **56.43**±0.80 |

# E    Additional results

**Expanded results on CLEVR and CLEVRTEX.**  In Table 2 we show expanded version of the results on CLEVR and CLEVRTEX benchmarks.

**Qualitative results on MOVi-A and MOVi-C.**  In  Fig. 1 and  2 we show additional qualitative results on MOVi-A and MOVi-C respectively. Following the results in main paper, the segments discovered by our method are semantically meaningful. Our object boundaries are of higher quality than the comparable methods. GWM suffers from oversegmentation of the objects. SCALOR has dificulty with complex datasets, such as MOVi-C as observed in Fig. 2. SAVI's object boundaries do not conform to object shape. We also provide additional failure cases of our model in Fig. 3. Our model has difficulty with objects that have complex motion for our affine formulation to model ably.

# Additional references

[1]  Christopher P Burgess, Loic Matthey, Nicholas Watters, Rishabh Kabra, Irina Higgins, Matt Botvinick, and Alexander Lerchner. Monet: Unsupervised scene decomposition and representation. *arXiv preprint arXiv:1901.11390*, 2019.

[2]  Subhabrata Choudhury, Laurynas Karazija, Iro Laina, Andrea Vedaldi, and Christian Rupprecht. Guess What Moves: Unsupervised video and image segmentation by anticipating motion. In *British Machine Vision Conference (BMVC)*, 2022.

[3]  Eric Crawford and Joelle Pineau. Spatially invariant unsupervised object detection with convolutional neural networks. In *Proceedings of the AAAI Conference on Artificial Intelligence*, volume 33, pages 3412–3420, 2019.

[4]  Patrick Emami, Pan He, Sanjay Ranka, and Anand Rangarajan. Efficient iterative amortized inference for learning symmetric and disentangled multi-object representations. In *Proceedings of the 38th International Conference on Machine Learning*, pages 2970–2981. PMLR, 2021.

[5]  Martin Engelcke, Oiwi Parker Jones, and Ingmar Posner. Genesis-v2: Inferring unordered object representations without iterative refinement. In *Advances in Neural Information Processing Systems*, volume 34, 2021.

[6]  Klaus Greff, Raphaël Lopez Kaufman, Rishabh Kabra, Nick Watters, Christopher Burgess, Daniel Zoran, Loic Matthey, Matthew Botvinick, and Alexander Lerchner. Multi-object representation learning with

---

[4]In our experiments, Y axis is pointing down and X is pointing right.

iterative variational inference. In *International Conference on Machine Learning*, pages 2424–2433. PMLR, 2019.

[7] Jindong Jiang and Sungjin Ahn. Generative neurosymbolic machines. In *Advances in Neural Information Processing Systems*, volume 33, pages 12572–12582, 2020.

[8] Jindong Jiang, Sepehr Janghorbani, Gerard De Melo, and Sungjin Ahn. Scalor: Generative world models with scalable object representations. In *International Conference on Learning Representations*, 2020.

[9] Laurynas Karazija, Iro Laina, and Christian Rupprecht. Clevrtex: A texture-rich benchmark for unsupervised multi-object segmentation. In *Thirty-fifth Conference on Neural Information Processing Systems Datasets and Benchmarks Track*, 2021.

[10] Thomas Kipf, Gamaleldin Fathy Elsayed, Aravindh Mahendran, Austin Stone, Sara Sabour, Georg Heigold, Rico Jonschkowski, Alexey Dosovitskiy, and Klaus Greff. Conditional object-centric learning from video. In *International Conference on Learning Representations*, 2022.

[11] Zhixuan Lin, Yi-Fu Wu, Skand Vishwanath Peri, Weihao Sun, Gautam Singh, Fei Deng, Jindong Jiang, and Sungjin Ahn. SPACE: Unsupervised object-oriented scene representation via spatial attention and decomposition. In *International Conference on Learning Representations*, 2020.

[12] Francesco Locatello, Dirk Weissenborn, Thomas Unterthiner, Aravindh Mahendran, Georg Heigold, Jakob Uszkoreit, Alexey Dosovitskiy, and Thomas Kipf. Object-centric learning with slot attention. In *Advances in Neural Information Processing Systems*, volume 33, pages 11525–11538, 2020.

[13] Tom Monnier, Elliot Vincent, Jean Ponce, and Mathieu Aubry. Unsupervised layered image decomposition into object prototypes. In *Proceedings of the IEEE/CVF International Conference on Computer Vision (ICCV)*, pages 8640–8650, 2021.

[14] Dmitriy Smirnov, Michael Gharbi, Matthew Fisher, Vitor Guizilini, Alexei A Efros, and Justin Solomon. Marionette: Self-supervised sprite learning. In *Advances in Neural Information Processing Systems*, volume 34, 2021.

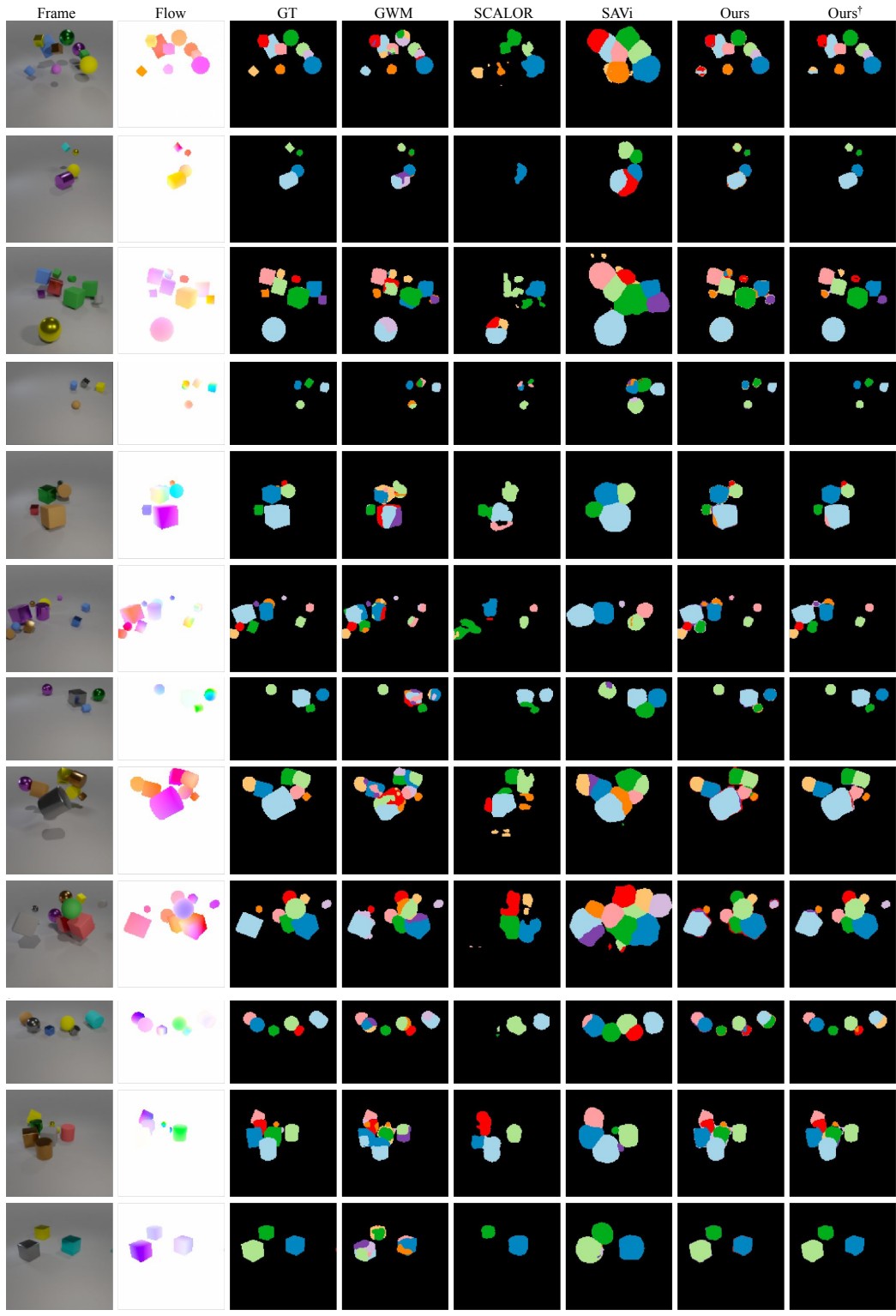

Figure 1: Additional qualitative comparison on MOVi-A. Our method performs consistently well compared to other methods. † – indicates post-processing.

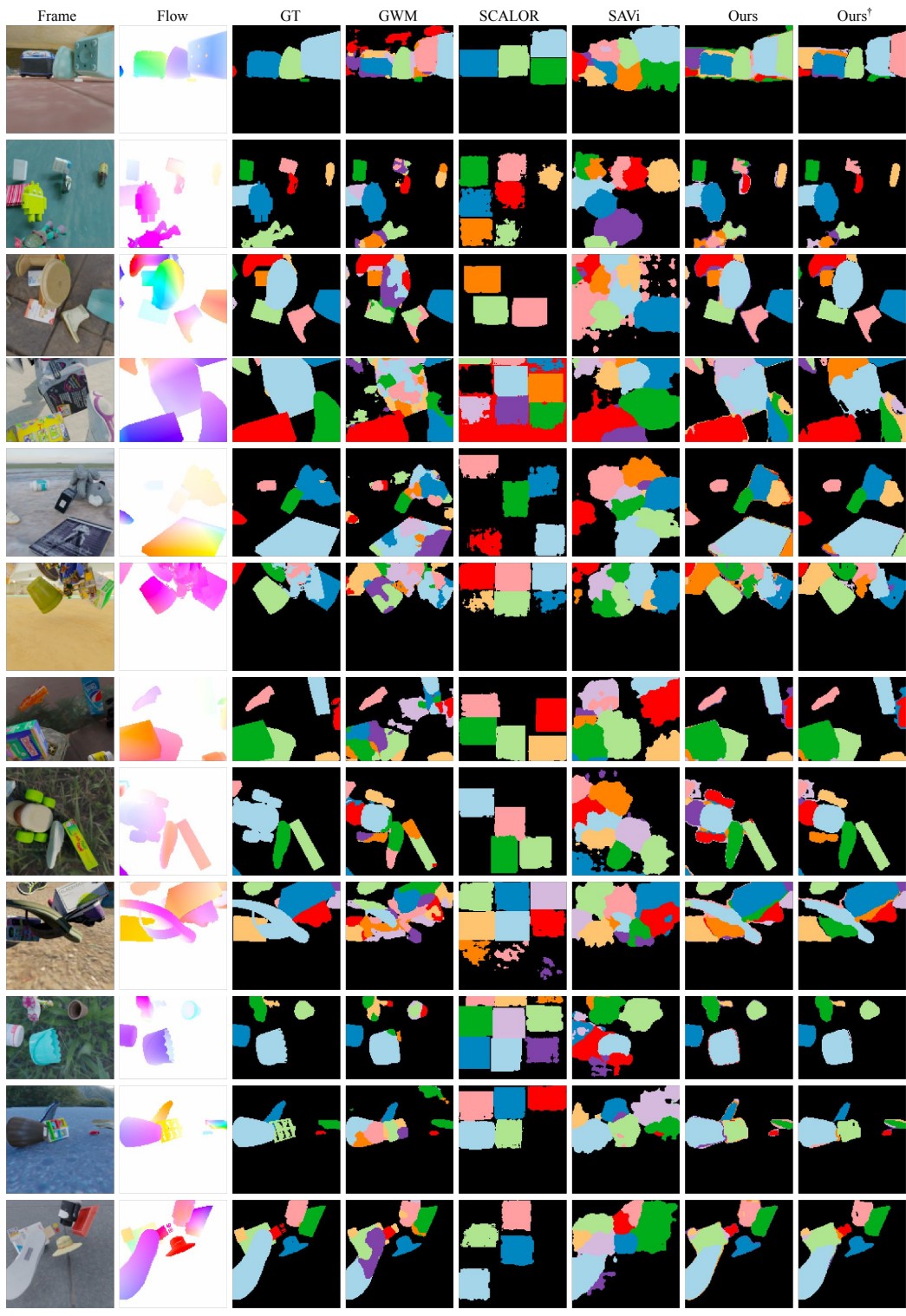

Figure 2: Additional qualitative comparison on MOVi-C. [†]– indicates post-processing.

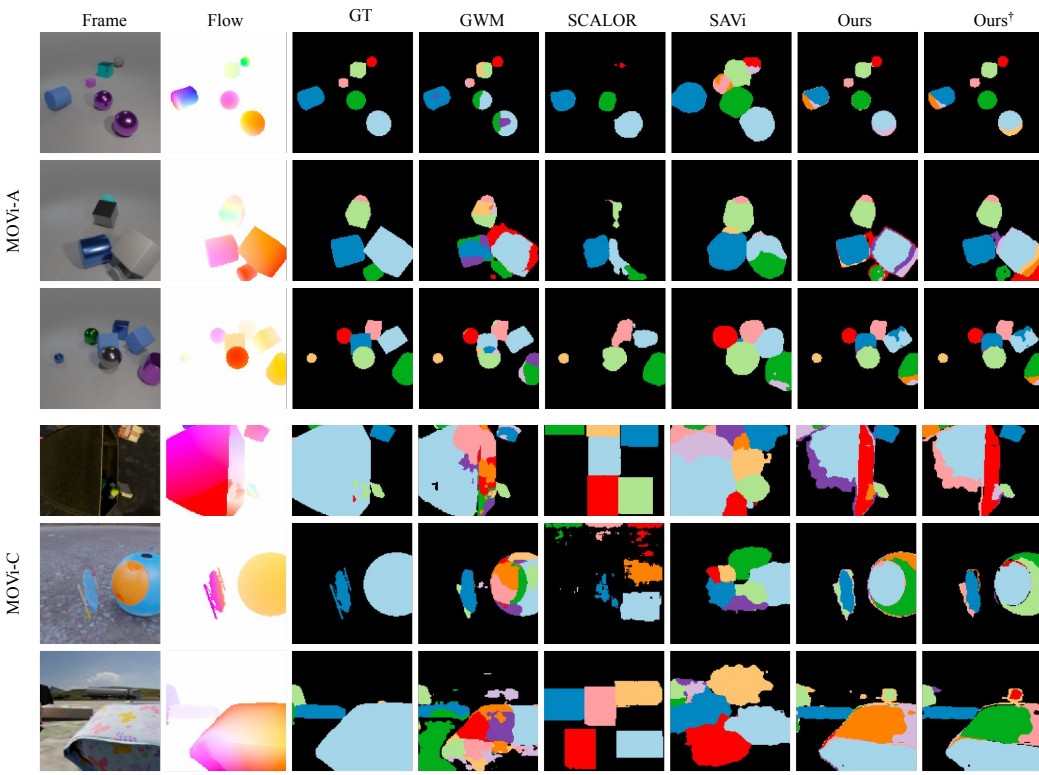

Figure 3: Additional examples of failure cases on MOVi-A and MOVi-C. [†]– indicates post-processing.