# OpenReview forum: "Unsupervised Multi-Object Segmentation by Predicting Probable Motion Patterns"
_NeurIPS.cc/2022/Conference — NeurIPS 2022 Accept_

### Official Review · Reviewer_bRe4 · 2022-07-07

**Rating:** 3
**Confidence:** 5
**Soundness:** 3 good
**Presentation:** 2 fair
**Contribution:** 2 fair

**Summary:**

The authors propose to learn unsupervised instance segmentation by grouping regions that tend to produce consistent optical flow. In particular, a network takes an image as input and predicts a distribution over k categories (segments) for every pixel. To compute the loss It also takes ground truth optical flow as input and a geometry-based, probabilistic objective then encourages the optical flow in each resulting region to conform to an affine motion model. The assumption is that each rigid object has a distinct affine motion pattern and thus the regions will correspond to object instances that move in that frame (not that the model itself only sees static frames, and learns the appearance of groups of pixels that tend to move together in the data). Crucially, this approach assumes that the camera in all videos is static, which the authors never explicitly mention. In the experiments on a few toy, synthetic datasets the proposed approach outperforms or performs on par with prior work, especially when evaluated on static images.

**Questions:**

Please evaluate your method on videos with a moving camera.

Please compare to a simpler baseline that computes motion segmentation with a sota method and uses it to supervise the masks.

Please compare to prior work using exactly the same backbone network.

How is k chosen? How robust is your approach to the choice of k?

**Limitations:**

The authors completely ignored the main limitation of their work - it's reliance on static camera.

**Strengths And Weaknesses:**

Strengths:

The intuition behind the approach is fairly straightforward and sound (grouping pixels in the optical flow field based on flow direction and magnitude does result in segmenting rigidly moving instances, it is the classic approach for motion segmentation).

The actual formulation of using affine motion model consistency to supervise a grouping method is novel to the best of my knowledge.

The paper is well written with almost no typos or grammatical mistakes.

The approach seems to work well in toy videos with a static camera.

A minimal ablation study is provided.


Weaknesses:

The authors don't discuss prior work on learning image-based, unsupervised object segmentation from motion. In particular, DyStaB of Yang et al., CVPR'21 which operates in a single object regime, and very recent Discovering Objects that Can Move of Bao et al., CVPR'22 which can segment multiple instances. The novelty of the proposed approach compared to these methods is in the loss formulation, but conceptually they are very close.

A major limitation that is not discussed in the paper is that the method relies on the static camera assumption. In fact, segmenting moving objects in the optical flow field with a static camera is trivial. Binary segmentation can be obtained by simple thresholding, but separating instances can also be done with a primitive approach, e.g. by fitting an affine motion model. Going to the more general setting of a moving camera is much harder, however, and would require a very different approach.

The objective is fairly complex and is not described in sufficient detail (both for the intuition behind the design choices and for the implementation details). In fact, it took me a while to figure out if the model actually predicts optical flow (or some proxy for it), or if the flow is only used to compute the objective function. The latter seems to be the case, making the title misleading. Implementation details are provided in the supplementary, showing that the objective is fairly brittle and requires different hyper-parameters on different datasets.

On the same note, the complexity of the proposed objective seems unnecessary and the authors do not compare to any simpler baselines. For example, a state-of-the-art motion segmentation algorithm could be used to extract moving object segments first and these segments could them be used to supervise the masks directly. The benefit of such a baseline, besides simplicity, is that it would allow the method to generalize to videos with a moving camera, since motion segmentation methods are designed to handle the most general setting.

The authors use a stronger backbone network compared to prior work (Mask2Former vs shallow CNNs used by most baselines), making all the experimental evaluations invalid. It is thus impossible to say if the proposed objective provides and real benefits compared to prior work.

---

> ### Author Response · Authors · 2022-08-02
> **Reply to Reviewer bRe4 [2/2]**
>
> > In fact, it took me a while to figure out if the model actually predicts optical flow (or some proxy for it), or if the flow is only used to compute the objective function. The latter seems to be the case, making the title misleading.
>
> We believe the title is quite apt. “Predicting <...> patterns” rather than the motion itself (which would be “predicting optical flow”) — because we task the network to carve out regions where certain patterns might be present. We included “probable motion” as we define a probability distribution of optical flow patterns which the model seeks to satisfy (make more probable predictions). This gives “Predicting probable motion patterns”. We are willing to change the title if there is consensus between the reviewers that this would improve clarity.
>
> > Implementation details are provided in the supplementary, showing that the objective is fairly brittle and requires different hyper-parameters on different datasets.
>
> This is not true. We show comparable results (within ~1.5%) with the _same_ hyperparameters for various datasets (Table 1(a), Supplementary Material). As part of our method, we also propose unsupervised approaches for the initial estimate of the covariance in a dataset-specific manner, that give slightly better performance, which we report in the main paper. However, the differences in performance are small and do not make or break the method.
>
> > On the same note, the complexity of the proposed objective seems unnecessary and the authors do not compare to any simpler baselines.
>
> We have a comparison with the recent proposed method by Choudhury et al. [5], primarily due to its closeness to our approach and ability to be expanded to a multi-object setting. As their approach is similar but simpler in its formulation (non-probabilistic), that comparison should answer the reviewer’s concern. We touch on the similarity and the benefits of our more principled formulation in the section A.2 of the supplementary material.
>
> > For example, a state-of-the-art motion segmentation algorithm could be used to extract moving object segments first and these segments could them be used to supervise the masks directly. The benefit of such a baseline, besides simplicity, is that it would allow the method to generalize to videos with a moving camera, since motion segmentation methods are designed to handle the most general setting.
> > Please compare to a simpler baseline that computes motion segmentation with a sota method and uses it to supervise the masks.
>
> The reviewer’s reference [Bao22] implements a more complex version of the suggested baseline, using masks to supervise a slot-attention model. As we have shown (global response), we outperform this approach.
>
> > The authors use a stronger backbone network compared to prior work (Mask2Former vs shallow CNNs used by most baselines), making all the experimental evaluations invalid. It is thus impossible to say if the proposed objective provides and real benefits compared to prior work.
> > Please compare to prior work using exactly the same backbone network.
>
> We disagree with the reviewer on this statement. The appendix (Table A.1) contains a comparison of backbones using UNet, ResNet50 and Swin-tiny with similar results between all variants.
>
> Additionally, many previous approaches have the motion model built into a custom network architecture (e.g. SAVi, SCALOR) and thus decoupling architecture and method is impossible. A benefit of our method is the ability to use an off-the-shelf segmentation network because the motion model is confined to the loss term.
>
> Furthemore, we also include experiments where our probabilistic approach is added to a previous method without any other changes (Table 3.c), improving its performance.
>
> To further address this concern, we test in the table below two more backbones, obtaining comparable results even with ResNet18. We also include an experiment where we use the MaskFormer architecture (_mf_) as in our other baseline GWM [5] (Table 1) showing better performance still.
>
> |                      |        | MOVi A |        | MOVi C |        | MovingClevr |
> |:-------------------- | ------:| ------:| ------:| ------:| ------:| -----------:|
> |                      | FG-ARI |   mIoU | FG-ARI |   mIoU | FG-ARI |        mIoU |
> | m2f (swin-tiny)      |  83.48 |  72.61 |  58.59 |  35.67 |  88.80 |       69.62 |
> | m2f (resnet50)       |  83.44 |  68.06 |  60.32 |  34.80 |  90.40 |       67.07 |
> | m2f (***resnet18***)|  84.04 |  67.48 |  60.84 |  35.69 |  90.31 |       67.33 |
> | ***mf*** (swin-tiny)|  81.78 |  71.28 |  54.45 |  33.67 |  71.07 |       51.06 |
>
> > How is k chosen?
>
> For direct comparison, $k$ is matched to the setting used in prior work [23, 24].

---

> > ### Comment · Reviewer_bRe4 · 2022-08-08
> > **Re:re**
> >
> > I thank the authors for their detailed response, which addressed most of my concerns. However, the fairness of comparison to the state-of-the-art remains a major issue. It is encouraging to see that the proposed approach is fairly robust to the backbone architecture, but the prior works which the authors claim to outperform use much more shallow backbones (6-layer CNN) than those reported above. For a fair comparison, exactly the same backbones need to be used in the state-of-the-art comparison table.
> >
> > The same goes for the new comparison to Bao et al. All the methods reported in that work are evaluated with a ResNet50 backbone. The authors need to use an equivalent variant of their model for comparison.

---

> > > ### Author Response · Authors · 2022-08-09
> > > **Authors Reply**
> > >
> > > We thank the reviewer for taking time to read our response.
> > >
> > > > I thank the authors for their detailed response, which addressed most of my concerns.
> > >
> > > We are happy to have addressed most of the concerns. We would be glad if the reviewer could raise their score to reflect this.
> > >
> > > We shall reply to other questions in the above response to our global comment.

---

> ### Author Response · Authors · 2022-08-02
> **Reply to Reviewer bRe4 [1/2]**
>
> We thank the reviewer for taking time to review our work. We would like to address some misunderstandings (especially the static camera assumption) and answer further questions/requests.
>
> > The authors don't discuss prior work on learning image-based, unsupervised object segmentation from motion. In particular, DyStaB of Yang et al., CVPR'21 which operates in a single object regime, and very recent Discovering Objects that Can Move of Bao et al., CVPR'22 which can segment multiple instances. The novelty of the proposed approach compared to these methods is in the loss formulation, but conceptually they are very close.
>
> There is indeed a vast literature on unsupervised segmentation, so we have focussed specifically on _multi-object_ decomposition from motion (e.g. SAVi) for a direct comparison to our work. As DyStaB performs foreground-background segmentation, it is not directly comparable to us. However, we will add a section to related work discussing motion-based single/salient object detection explaining the differences.
>
> Although the concurrent work “Discovering Objects that Can Move”, Bao et al. (CVPR 2022), was published after the submission deadline, for completeness, we have run additional experiments following their setting to compare to their approach. We discuss this in detail in a global response to all reviewers, including quantitative comparisons on real-world data. We show that we outperform [Bao22]  as well as previous work, and we will include these experiments in the final version of our paper.
>
> > A major limitation that is not discussed in the paper is that the method relies on the static camera assumption. In fact, segmenting moving objects in the optical flow field with a static camera is trivial. Binary segmentation can be obtained by simple thresholding, but separating instances can also be done with a primitive approach, e.g. by fitting an affine motion model.
> > Going to the more general setting of a moving camera is much harder, however, and would require a very different approach.
> > Please evaluate your method on videos with a moving camera.
>
> We ***do not***, in fact, assume that the camera is static. Instead, the model of each region’s motion (affine) accounts for the ***combined*** effect of camera motion _and_ object motion. The reviewer’s suggested baseline (segmentation by fitting an affine model) is a simplified version of one of our baselines, Choudhury et al. [5]. We achieve superior results (Table 2) and provide discussion and analysis as to why our proposed method is able to better handle multiple-object scenarios in Section A.2.
>
> To further emphasize the fact that our approach does _not_ rely on a static camera assumption, we report results on MOVi-D and MOVi-E datasets below. These variants of the MOVi dataset have an explicitly moving camera. This does not cause our method to under-perform in any way; we show equally strong results while using the ***same*** settings as for all other experiments.
>
> |                         |          | MOVi_D |          | MOVi_E |
> |:----------------------- | --------:| ------:| --------:| ------:|
> |                         |   FG-ARI |   mIoU |   FG-ARI |   mIoU |
> | SAVi [24] (+Bbox sup.)* | <45 |        | <55 |        |
> | Our (Simplified Cov.)   |    56.02 |  24.10 |    66.29 |  27.25 |
>
> (*- SAVi results from plot in [Singh22])
>
> As mentioned before, we also include results of our method applied to the real-world driving dataset KITTI, which naturally has strong ego-motion. On this challenging setting, we also achieve state-of-the-art performance.
>
> > The objective is fairly complex and is not described in sufficient detail (both for the intuition behind the design choices and for the implementation details).
>
> We will further detail our expanded explanation in the supplementary material to bring additional clarity to the readers.
> While the reviewer may perceive the model “fairly complex”, the math is directly derived from a small and simple set of assumptions  (Section 3, Sec. A). We believe our method is quite simple in practice (Eq. 1). It only requires evaluating the loss on the output of the segmentation network and the flow. This is more straightforward than a pipeline using one method to obtain noisy labels and training a new network.
>
> As an example, the proposal of Bao et al. [Bao22] centers around a way of including motion segmentation soft labels into a slot-attention network. This is an architecture-specific approach that relies on a different method to obtain soft labels. In contrast, our approach is simple enough that it can be applied to different architectures (Table 1(a) in Supp. Material) and even added to existing methods to improve their performance, as we show in our ablation experiments (Table 3(c)).

---

### Official Review · Reviewer_9r13 · 2022-07-11

**Rating:** 8
**Confidence:** 4
**Soundness:** 3 good
**Presentation:** 4 excellent
**Contribution:** 3 good

**Summary:**

The authors approach unsupervised instance segmentation of images by exploiting motion
patterns in videos during training. Given an RGB input frame, a neural network is trained
to predict a segmentation that allows to approximate the optical flow field of that frame
well when using a simple, parametric motion pattern for each segment (e.g., affine
motion). Since the segmentation is predicted based on the RGB frame only, the network
is trained to predict segments that *potentially* move and can be used to segment images
after training. On recent, synthetic multi-object datasets the proposed model is
shown to outperform recent state-of-the-art object-centric models in almost all cases.

**Questions:**

- Are the appearance-based models also trained on frames of the video variants of
  CLEVR and CLEVRTex? Or directly on CLEVR/CLEVRTex? In the latter case, an experiment
  testing whether the different training data influences the performance of appearance
  based models should be included.

- Details about the dataset generation are not contained in the supplement,
  although promised in the main paper.


**Limitations:**

## Limitations
The authors discuss limitations of their method adequately in a separate section. One
limitation to add in my view would be the necessity of ground truth optical flow. The
ablation study has shown a substantial performance drop when using optical flow
predicted by SMURF, which probably makes it difficult to apply the method to natural
videos where ground truth optical flow is not available.


**Strengths And Weaknesses:**

Unsupervised segmentation is an active area of research. The general approach of
defining objects as regions with a simple motion pattern when they move is not new; the
respective related work is cited. The particular instantiation of the
approach in this work however is novel and shown to clearly improve over previous models in
almost all cases. A particular strength of the paper is the thorough comparison with
previous models. The authors tested a large range of models on the datasets using
multiple seeds, going beyond most related papers in the field.

On the other hand, the authors only compare to object-centric models that not only aim
at learning to segment images or videos but also to learn a useful representation of
those objects. Another line of works exists that only focusses on unsupervised
segmentation, with models that often make use of optical flow for training (and
sometimes inference) and can be applied to more realistic data. The authors
cite respective works, e.g. Mahendran et al. 2018, Yang et al. 2019, Choudhury et al. 2022.
Empirical comparisons with these models would have been informative and would strengthen
the paper, as these would allow to more directly compare the different approaches for
optical flow based unsupervised segmentation.

---

> ### Author Response · Authors · 2022-08-02
> **Reply to Reviewer 9r13**
>
> We thank the reviewer for the encouragingly positive review and assessment of our contributions and presentation. We address the comments and questions below.
>
> > Another line of works exists that only focusses on unsupervised segmentation, with models that often make use of optical flow for training (and sometimes inference) and can be applied to more realistic data. The authors cite respective works, e.g. Mahendran et al. 2018, Yang et al. 2019, Choudhury et al. 2022. Empirical comparisons with these models would have been informative and would strengthen the paper, as these would allow to more directly compare the different approaches for optical flow based unsupervised segmentation.
>
> Our objective is to perform multi-object segmentation. Many unsupervised segmentation methods operate only in foreground/background scenarios, with extensions to the multi-object case unclear or non-trivial. We compare (Tab. 2) with the proposal of Choudhury et al. [5] (where the extension to multi-object is relatively straight-forward) and show better results. Based on your suggestion, we now also conducted experiments on real-word data (please see the global post above), where we achieve state-of-the-art performance on a real-world dataset.
>
> > Are the appearance-based models also trained on frames of the video variants of CLEVR and CLEVRTex? Or directly on CLEVR/CLEVRTex? In the latter case, an experiment testing whether the different training data influences the performance of appearance based models should be included.
>
> The appearance-based models are trained on the original CLEVR/CLEVRTex using implementations/settings provided by Karazija et al. [23]. Below, we include a revised version of Table 3(c) with GWM trained on our video variant of the dataset.
>
> |                               |        | ClevrTex |        |   OOD |        |  Camo |
> |:----------------------------- | ------:| --------:| ------:| -----:| ------:| -----:|
> | Model                         | FG-ARI |     mIoU | FG-ARI |  mIoU | FG-ARI |  mIoU |
> | GNM [19] (ClevrTex)           |  53.38 |    44.39 |  48.44 | 42.87 |  15.72 | 18.53 |
> | GNM [19] (MovingClevrTex)     |  18.01 |    31.47 |  15.57 | 30.23 |   0.21 | 14.68 |
> | GNM+Our Loss (MovingClevrTex) |  63.84 |    55.26 |  59.01 | 48.65 |  51.00 | 47.63 |
>
> In general, we observe that the method under-performs when trained on the video data, likely due to lack of scene diversity (to keep the dataset the same size, we trade-off the number of different scenes with video frames). We therefore already compared to a stronger baseline using the original (static) CLEVR/CLEVRTex training data. Nonetheless, augmenting the method with our loss, consistently improves performance.
>
> > Details about the dataset generation are not contained in the supplement, although promised in the main paper.
>
> Thank you for pointing this out! We will update the supplementary with the missing section. In brief, we rely on the dataset generation code of Karazija et al. [23]. We augmented the code to (1) render multiple frames for each scene, (2) enable the built-in physics simulation of Blender (3) and sample initial velocity (normal and angular) for either one, two or all objects in the scene. The directions of the velocity are constrained to roughly aim at the scene center to maintain all objects in the frame and enable complex interactions such as collisions and occlusions. All other settings are the same as in [23]. The modified generation code is included in the original submission.
>
> > One limitation to add in my view would be the necessity of ground truth optical flow. The ablation study has shown a substantial performance drop when using optical flow predicted by SMURF, which probably makes it difficult to apply the method to natural videos where ground truth optical flow is not available.
>
> We thank the reviewer for the suggestion. We shall include a mention of the limitation that our method currently does not directly model complex noise likely present in real-world scenarios and would benefit from direct handling of this aspect in the future.
> We note, however, we also discuss our findings on real-world data in the global response to all reviewers (please see above). In these experiments, we follow the setting of [Bao22]., which uses KITTI data and RAFT as the flow estimator. To explicitly account for noisy flow areas in real-world data, we found it beneficial to include an additional loss term (dubbed _Warp Loss_), which adds more than 10 percentage points to the FG-ARI performance of our model. We describe this term in detail as part of our global response.
> This is a simple way that the model could be made robust towards real-world flow estimators.

---

> > ### Comment · Reviewer_9r13 · 2022-08-10
> > **Re: Reply to Reviewer 9r13**
> >
> > Thank you for your detailed response. The additional results on real-world data look indeed very promising. I would like to encourage you to extend the limitations and conclusion section regarding scaling to real-world data with reference to the new results.
> >
> > Overall my concerns have been well addressed. I am keeping my rating of strongly recommending this paper to be accepted.

---

### Official Review · Reviewer_iFdZ · 2022-07-19

**Rating:** 7
**Confidence:** 3
**Soundness:** 3 good
**Presentation:** 2 fair
**Contribution:** 3 good

**Summary:**

The authors train a segmentation network that processes still images to produce object masks. Video based supervision is used. The masks predicted for a single image are converted to optical flow using a motion model proposed by the authors. Two different motion models are considered. Hand-crafted post-processing is used to improve final segmentation performance.

The key novelty is the proposed approach for matching object masks to an optical flow representation. The authors use a motion model (i.e. translation only / affine transform on each object region), introduce an efficient and differentiable implementation of matching, and evaluate the improvements over multiple datasets. Notable performance improvements are on out-of-domain datasets. Also, the method scores relatively better on the mIoU metric, denoting the multi-object capability.

Ablations highlight the significance of ground truth optical flow for training (low performance without it) as well as better motion model (translation only gives lower results).

The authors include code in supplementary, and promise to share all code, datasets publicly in future.

**Questions:**

1. Line 114: more explanation for reasoning in ELBO for regularization
2. A direct formulation of the final loss function

**Limitations:**

Yes, limitations discussed well.

**Strengths And Weaknesses:**

### Strengths
1. A novel motion model based matching of object masks to optical flow
2. The authors obtain considerable improvements over a range of datasets

### Weaknesses
1. Using ground-truth optical flow seems crucial for good performance (Table 3.a) - is this method scalable to real world data? Other methods using video supervision can operate on real-world data (e.g. [1])
2. Is the method really benefiting from the motion model or simply learning segmentation from the high quality ground-truth optical flow masks?
3. The authors note how similar models struggle in complex settings, but limit all their evaluations to synthetic datasets: how would this translate to real world settings?
4. Lack of clarity/simplicity in final loss function: can the authors construct a simpler version of the final loss between masks output from model and optical flow labels (closer to the actual implementation in code)? Many terms in the given equations are constants (or are they learnable params?)
5. Is comparing to other methods with the additional post-processing fair? The authors do include results without post-processing that are still above SOTA; those may be a better comparison.

[1] [Discovering Objects that Can Move](https://openaccess.thecvf.com/content/CVPR2022/papers/Bao_Discovering_Objects_That_Can_Move_CVPR_2022_paper.pdf)

---

> ### Author Response · Authors · 2022-08-02
> **Reply to Reviewer iFdZ [2/2]**
>
> > Lack of clarity/simplicity in final loss function: can the authors construct a simpler version of the final loss between masks output from model and optical flow labels (closer to the actual implementation in code)? Many terms in the given equations are constants (or are they learnable params?)
>
> Our loss formulation is grounded in probabilistically modeling flow fields using a motion prior. The provided formulation of the loss is what is implemented in code (please see the provided code _code/src/dist.py:141_). Note that there is only a single term that is constant ($HW \log 2 \pi \sigma^2$) and no learnable parameters. The constant term maintains a normalized probability distribution. All other terms depend on either the flow or the predicted masks. As we discuss in the supplementary (A.2 Further justification), these terms end up playing a role in for example biasing the model to use fewer masks to explain the objects.
>
> > Is comparing to other methods with the additional post-processing fair? The authors do include results without post-processing that are still above SOTA; those may be a better comparison.
>
> Thanks. We included both results for this reason. Note that the post-processing in question is a simple deterministic hyper-parameter-free connected-components processing of the output masks that could be directly included in the model. The cost of this is negligible. As we state Section 3 (L175-182), the post-processing is included to address the shortcoming of the method where parts of the scene could be grouped in the same region for coincidentally having similar motion (e.g., due to free falling). As other methods could potentially benefit from such filtering as well, we also applied the same post-processing to other methods to verify the results. However as we show in Table 1, prior work does not benefit from this post-processing universally and in fact is hurt by it in some cases.
>
> > Line 114: more explanation for reasoning in ELBO for regularization
>
> Our loss is evaluated based on the sampled masks rather than the underlying (mask) distribution. The regularization term here aids in controlling the “shape’’ of the predicted mask distribution. While in deriving the loss we leave an option to adopt a better/complex mask distribution prior, in our experiments we simply use a uniform prior. We include the common $\beta$ parameter from constraint optimisation [Higgins16] as that enables us to encourage the network to predict more solid masks by annealing this value, as discussed in the supplementary material (Sec. B, L80-84).
>
> > A direct formulation of the final loss function
>
> The direct formulation of the final loss function would be rather lengthy and tedious as it would require manually performing many simple linear algebra manipulations that numeric processing libraries readily make available (e.g., finding the determinant of an inverse of a 3x3 matrix). Also implementing such a form would be wasteful as many terms can be computed jointly using efficient vectorised implementations. We describe in the supplementary material (Sec. A.1 “Affine motion likelihood”) the exact equations and steps how the loss is translated to code to make use of available BLAS libraries.
> We will further update this section with additional explanation, the above discussions, and pseudo code showing how the provided equations can be combined to evaluate the loss.
>
> ### References
>
> [Higgins16] - Higgins, Irina, et al. "[Beta-vae: Learning basic visual concepts with a constrained variational framework.](https://openreview.net/forum?id=Sy2fzU9gl)"  ICLR‘16.

---

> > ### Comment · Reviewer_iFdZ · 2022-08-08
> > **Reviewer Feedback**
> >
> > Thank you for the detailed responses addressing most concerns raised in reviews.
> >
> > The results on KITTI are particularly interesting and helpful to understand the performance of proposed method on real world data. The use dataset specific hand-crafted priors in ego-motion conv questions the generality of the method. The warp-loss is an interesting addition that integrates a general prior regarding videos to the loss. Overall, this is a great addition to the paper.
> >
> > The concern on ground truth "optical flow masks" was similar to the concerns of another reviewer. In many synthetic datasets, optical flow segments the objects present quite well - in fact they can be comparable to the actual segmentation ground-truth. However, the KITTI results (where dataset contains camera motion) and especially the improvements with the "warp-loss" component clarify this.
> >
> > The explanations relevant to the loss function / regularization is clarifies the concerns.
> >
> > Overall I feel the paper is fit for acceptance.

---

> ### Author Response · Authors · 2022-08-02
> **Reply to Reviewer iFdZ [1/2]**
>
> We thank the reviewer for taking time to review our work and favorably assessing contributions, soundness and presentation. We would like to address some points in the weaknesses section and answer the questions raised.
>
> > Using ground-truth optical flow seems crucial for good performance (Table 3.a) - is this method scalable to real world data? Other methods using video supervision can operate on real-world data (e.g. [1])
> >The authors note how similar models struggle in complex settings, but limit all their evaluations to synthetic datasets: how would this translate to real world settings?
>
> As a similar concern has been raised by other reviewers, we have created a global response (please see [above](https://openreview.net/forum?id=_w2-1nXNjvv&noteId=atbG9TkI4MRU)) to discuss how our method translates to real-world settings. We conduct additional experiments on the KITTI dataset and show that our method still outperforms prior and concurrent work on this more complex, real-world setting.
>
> > Is the method really benefiting from the motion model or simply learning segmentation from the high quality ground-truth optical flow masks?
>
> We are not entirely sure what the reviewer means by optical flow masks. The optical flow is not masked. Our network is trained to output segmentation masks. However, no form of annotations is used for training, only for evaluation. The loss is derived based on how well the predicted masks carve up the image into coherent optical flow regions, and _not_ computed against ground truth masks.
> The motion model here is used explicitly to enable learning segmentation from the optical flow (HxW 2-channel image), rather than directly reconstructing the flow. It enables defining a prior that encodes a notion that the induced optical flow distribution comes from underlying objects. As we discussed in Section A.2, (Appendix), the probabilistic treatment of the motion model enables additional benefits in the multi-object paradigm that prior work lacks. As we show in our evaluation (Table 2), our method also compares favorably to methods that do not include motion models but rely on predicting optical flow directly (SAVi).

---

### Official Review · Reviewer_9Fas · 2022-07-20

**Rating:** 7
**Confidence:** 3
**Soundness:** 3 good
**Presentation:** 4 excellent
**Contribution:** 4 excellent

**Summary:**

The author presents a method for unsupervised object segmentation by using temporal motion patterns. The paper transforms the segmentation as a minimizing ELBO problem. Some practical tricks of Gumbel-softmax have been applied to make the ELBO differentiable. The evaluations on the physics-related simulation datasets were conducted and outperforms the other baselines.

**Questions:**

One question is how well can the methods perform in real-world cases. I’m not asking about adding more results in the rebuttal. It would be helpful if the authors can explain if the methods can apply to real-world cases and what are the potential limitations. For example, the KITTI dataset is one possible option, as it involves mainly rigid motions. But meanwhile, KITTI is a moving camera case, what are the possible concerns if the model is applied to a moving camera case.

How did the prior distribution p0(m) initialize and how did different prior distributions affect the results?

It will be interesting to see what the authors’ thoughts on the potential to apply the proposed methods to the no-rigid motion are? What are the further requirements to achieve that?

Some other detailed comments:

Line 111: Please elaborate on the assumption “We then assume that the flow depends only the regions”

Line 138: Please justify that “ We assume that regions are statistically independent ” is a reasonable assumption.

Matrix I first appeared in equation (5) but is explained in line 161.

line 2 form -> from.

Ling 156, missing space after as.


**Limitations:**

As the author stated, sometimes the motion doesn't provide enough information to segment the object. Maybe adding scene flow (3d flow with depth) can help to improve the performance.

**Strengths And Weaknesses:**

This paper converts a classic optimization problem into a differentiable learning problem, the overall idea is very interesting. The experiments have shown the proposed method has achieved the SOTA performance on simulated datasets.

One weakness is no real-world scenario experiments, it would be more convincing if some of the experiments were conducted on real-world videos.

---

> ### Author Response · Authors · 2022-08-02
> **Reply to Reviewer 9Fas**
>
> We thank the reviewer for taking the time to read our work and positively assess its soundness, presentation and contributions. We would like to address the questions and concerns raised in the review.
>
> > One question is how well can the methods perform in real-world cases.
>
> Thanks! As some of the other reviewers had a similar question, we have created a global response (please see [above](https://openreview.net/forum?id=_w2-1nXNjvv&noteId=atbG9TkI4MRU)) to address this. It turns out it does work: we have conducted additional experiments on _real-world_ data demonstrating that our method outperforms prior and concurrent work.
>
> > How did the prior distribution p0(m) initialize and how did different prior distributions affect the results?
>
> The prior distribution $p_0(m)$ is a uniform categorical distribution, meaning $p_0(m) = 1/k$, where $k$ is the number of components. This distribution is parameter-free.
> During experimentation, we also considered a different mask prior distribution, a 2D Gaussian Mixture Model (with K components) by considering centroids and pixel coordinates as random variables to encode object compactness. However, this did not result in a discernible difference in our experiments. Instead, we found that restricting the entropy of the learned mask distribution, which in practice meant that the mask output was more “peaky”, had a larger effect on performance. This was achieved by controlling $\beta$ parameter (Eq. 1) (See Supp. Mat. B, L80-85 for details).
>
> > It will be interesting to see what the authors’ thoughts on the potential to apply the proposed methods to the no-rigid motion are? What are the further requirements to achieve that?
>
> It largely depends on the contents and the extent of the non-rigid motions. Our model assumes a rigid-motion prior but does not necessitate working on only rigid motions. Small amount of non-rigidity can be treated as noise, much like the imperfect affine approximation of the flow field, and could be handled as is.
> For pronounced non-rigid motion (e.g., humans dancing or animals running), the motion could be explained at the level of object parts, thus a hierarchical segmentation scheme could be used to get around this and other problems common in this setting, such as self-occlusion and depth discontinuities, as we briefly mention in Sec. 4.5 (L274-275).
> We will add this discussion to the future work section.
>
> > Line 111: Please elaborate on the assumption “We then assume that the flow depends only the regions”
>
> We assume that optical flow within a region depends only on the region itself and the other regions have no influence. Intuitively, this is a reasonable assumption, as in large, the movement/flow of an object does not depend on the background or other objects thus enforcing this assumption encourages regions to correspond to objects.
>
> > Line 138: Please justify that “ We assume that regions are statistically independent ” is a reasonable assumption.
>
> This follows the same reasoning as above. We assume that regions are statistically independent given their segmentations, that is the optical flow for the object depends only on the region. This in practice may confound two separate sources of motion for an object, e.g. camera egomotion and the objects’ inherent motion, however our model attempts to explain any such motion jointly for each object rather than disentangling possible sources of motions. Such a simplifying assumption has several benefits. Firstly, this helps induce a tractable probability distribution for an observed region. Secondly, this results in smaller regions being explained independently, which should align better with the affine motion prior assumption.
> We will include these explanations in the paper.
>
> > Typos and small errors.
>
> Thank you! We will address these in the revision.

---

> > ### Comment · Reviewer_9Fas · 2022-08-09
> > **Reply to authors**
> >
> > Thanks to the authors for providing detailed answers. The additional experiments on KIITI really helped to see the algorithm works in a real-world scenario, which resolved my main concern.  I hope there will be enough space to add those results to the main draft, which is important to the overall story.
> >
> > My other questions have been clarified and I don't have further questions. Overall I will keep my rating and recommend an accept.

---

### Author Response · Authors · 2022-08-02
**Global Response: Experiments on Real-World Data**

We thank all reviewers for their feedback and comments on the paper. For this rebuttal we have run additional experiments using real data as requested by reviewers which we collect here, and will refer to below in the individual responses.
We apply our method to the real-world scenario following the setting of the concurrent work by Bao et al. [Bao22]. Namely, we apply our method to the KITTI dataset using optical flow estimated by RAFT. Our findings are in the table below.

|                        | KITTI (FG-ARI) |
| ---------------------- | --------------:|
| SA [32]                |           13.8 |
| S-IODINE [14]          |           14.4 |
| MONet [3]              |           14.9 |
| SCALOR [20]            |           21.1 |
| MCG [Arbeláez14]       |           40.9 |
| Bao et al. [Bao22]     |           47.1 |
| Ours (Simplified Cov.) |           46.1 |
| Ours (Egomotion Cov.)  |     ***47.5*** |
| Ours (+Warp loss)      |           57.8 |

_Ours (Simplified Cov.)_ refers to our method trained out-of-the-box on KITTI using the simplified hyper-parameters used in our previous experiments (Sec. B of Supplementary Material). It already achieves comparable performance to Bao et al.. _Ours (Egomotion Cov.)_ refers to our method applied with the covariance for the motion prior distribution ($\Sigma$, Eq. 5) adjusted to account for strong egomotion characteristic of driving videos – we simply assume some positive scaling and shear in Y direction. This outperforms previous approaches and achieves state-of-the-art results.
Finally, we conduct experiments with an additional loss term (_Warp Loss_) to further aid in dealing with the noise of the estimated flow. This term simply enforces consistency between adjacent frames by warping the predicted masks using optical flow, as follows:

$L_{warp}(m_1, m_2, I_1, I_2, f_1, b_2) = w(I_2, f_1(I_1)) \cdot d(m_2, f_1(m_1)) + w(I_1, b_1(I_2)) \cdot d(m_1, b_2(m_2)),$

$w(I_a, I_b) = |I_a - I_b| / \sum_x (|I_a - I_b|)_x,$

$d(m_a, m_b) = D_{KL}(m_a || m_b)/2 + D_{KL}(m_b || m_a)/2,$

Where $m_1,m_2$ are predicted k-way categorical mask probabilities, $I_1,I_2$ are frames, $f_1(\cdot)$ indicates warping by forwards optical flow $f_1$ (or backward $b_2$). Regions where warping of RGB produces small errors will enforce matching of mask distributions.
Although not necessary to surpass state-of-the-art performance, we can see that our method with this additional term (_Ours (+Warp loss)_) performs even better and shows how a noisy flow estimator can be integrated in our method.

Overall, our method shows very strong performance in real world scenarios, improving the performance over previous and concurrent results.

### References

[Bao22] - Bao, Zhipeng, et al. "[Discovering Objects that Can Move.](https://openaccess.thecvf.com/content/CVPR2022/papers/Bao_Discovering_Objects_That_Can_Move_CVPR_2022_paper.pdf)", CVPR’22 (concurrent)

[Arbeláez14] - Arbeláez, Pablo, et al. "[Multiscale combinatorial grouping.](https://openaccess.thecvf.com/content_cvpr_2014/papers/Arbelaez_Multiscale_Combinatorial_Grouping_2014_CVPR_paper.pdf)" CVPR’14.

[Singh22] - Singh, Gautam et al. "[Simple Unsupervised Object-Centric Learning for Complex and Naturalistic Videos.](https://arxiv.org/abs/2205.14065)"

---

> ### Comment · Reviewer_bRe4 · 2022-08-08
> **Questions on the new comparison**
>
> I thank the authors for reporting these encouraging results. However, the question of fairness of the evaluation remain open here as well. All the methods in Bao et al. used a ResNet50 backbone. Please report the results of your method with exactly the same backbone.
>
> As to the warping loss, is it unique to the proposed approach, or can it be added to any learning-based method that predicts soft masks (egg, Bao et al)? If it can, then the corresponding results should be added as well.

---

> > ### Author Response · Authors · 2022-08-09
> > **Authors Answer**
> >
> > We thank the reviewer for the response.
> >
> > > Please report the results of your method with exactly the same backbone.
> >
> > Due to lack of time we are not able to do more additional experiments. However, we have already reported performance with different backbones and segmentation networks (Sup. Mat. - Table 1a, and expanded version below) showing our method is robust to work with many off-the-shelf segmentation networks.
> >
> > Importantly, we would like to draw attention to the fact that our model with a transformer architecture (SWIN-tiny) and ResNet-50 backbones are close in the parameter count and performance.
> >
> > Additionally, we are computationally more efficient than both SAVi and Bao et al. Our model requires a single GPU for 2 days (peak VRAM of 18Gb, 23Gb for KITTI), whereas SAVi requires 8 GPUs for 12 or 30 hours and Bao uses 4 GPUs for 3 days ([link](https://github.com/zpbao/Discovery_Obj_Move/issues/3)). This is because our method does not need computationally expensive custom architectures, and works well with only a segmentation network.
> >
> > We’ll update the tables with the additional requested experiments when they are ready.
> >
> > > can it be added to any learning-based method that predicts soft masks (egg, Bao et al)?
> >
> > Please note we improve over Bao et al. even without the mask warping loss. We included it as a simple way to deal with noisy flow estimates on real data. The warping loss, like our proposed method for flow coherence loss, can be applied to any learning-based method that predicts masks (incl. Bao et al., SAVi etc). We shall expand our table 3(c) carrying out these experiments for the final version, to include additional methods.

---

> > > ### Comment · Reviewer_bRe4 · 2022-08-09
> > > **Re:re**
> > >
> > > I appreciate the fact that the mode is computationally efficient and has a comparable number of parameters to ResNet50, but you cannot make claims about perfromance advantages over prior work unless the same backbone is used in evaluation (in particular, given how narrow the margin is). This is even more important for the results in the main paper, where prior works use clearly inferior backbones (e.g. 6-layer CNNs). This remains a major issue, making evaluating the advantages of the proposed approach over prior work very challenging.

---

### Meta-Review · Area_Chair_9NKz · 2022-08-31

**Recommendation:** Accept
**Confidence:** Less certain

**Metareview:**

This paper presents an approach for unsupervised multi-object segmentation. The majority of the reviewers believe the paper contains interesting technical materials that warrants its acceptance. The (only) remaining concern is from Reviewer bRe4, pointing out that the paper uses a more advanced backbone than the baselines. Although the other reviewers also agree to this point, they believe the ablations in the paper are valid and they justify the benefits of the approach. Overall, the ACs recommend the acceptance of the paper.

**Award:**

No

---

### Decision · Program_Chairs · 2022-09-14

Accept